# Avian Influenza Clade 2.3.4.4b: Global Impact and Summary Analysis of Vaccine Trials

**DOI:** 10.3390/vaccines13050453

**Published:** 2025-04-24

**Authors:** László Kovács, Máté Farkas, Péter Ferenc Dobra, Georgia Lennon, László Péter Könyves, Miklós Rusvai

**Affiliations:** 1Department of Animal Hygiene, Herd Health and Mobile Clinic, University of Veterinary Medicine, 1078 Budapest, Hungary; kovacs.laszlo@univet.hu (L.K.); georgia.lennon1998@gmail.com (G.L.); konyves.laszlo@univet.hu (L.P.K.); 2Poultry-Care Kft., 5052 Újszász, Hungary; 3National Laboratory of Infectious Animal Diseases, Antimicrobial Resistance, Veterinary Public Health and Food Chain Safety, University of Veterinary Medicine, 1078 Budapest, Hungary; 4Institute of Food Chain Science, Department of Digital Food Science, University of Veterinary Medicine, 1078 Budapest, Hungary; 5Department of Pathology, University of Veterinary Medicine, 1078 Budapest, Hungary; 6Vet-Diagnostics Kft., 5000 Szolnok, Hungary; rusvai.miklos@gmail.com

**Keywords:** highly pathogenic avian influenza, H5N1, vaccination, One Health, epidemic, animal welfare

## Abstract

Background: Avian influenza (AI), caused by orthomyxoviruses, is a globally significant disease affecting avian and non-avian species. It manifests in two variants, according to the two biovariants of the virus differentiated as highly pathogenic avian influenza (HPAI) and low pathogenic avian influenza (LPAI) strains, both of which compromise animal welfare, reduce productivity, and cause substantial economic loss. The zoonotic potential of HPAI strains, particularly the currently dominant clade 2.3.4.4b, raises concerns about public health and epidemic risks. This review assesses the results of current vaccine trials targeting HPAI clade 2.3.4.4b, emphasizing these studies because most outbreak strains in domestic poultry currently belong to this dominant clade. Methods: Multiple scientific databases comprised reports of research trials on vaccine efficacy against HPAI clade 2.3.4.4b. The Boolean term “Clade 2.3.4.4b AND vaccine” was entered into the following databases: PubMed, PubAg, Scopus, Cochrane Library, and ScienceDirect. Results: The resulting papers were analyzed. Studies revealed that antigenic similarity between vaccine and field strains enhances protective efficacy (PE), reduces viral shedding, and improves hemagglutination inhibition titers. While multivalent vaccines showed potential, results were inconsistent and varied depending on strain compatibility. Single-dose vaccines may provide sufficient PE for poultry, though ducks and geese often require multiple doses, and long-term PE is yet unknown. It was discovered that vector vaccines can provide appropriate PE against clade 2.3.4.4.b. Conclusions: Further analysis is needed as their effects may be short-lived, and subsequent doses may be required. Limited research exists on the long-term efficacy of these vaccines and their effectiveness in many avian species. Addressing these gaps is crucial for optimizing vaccination strategies. A re-evaluation of vaccination strategies is recommended but essential to implement adequate biosecurity measures on in poultry farms. This review synthesizes current evidence and may assist veterinarians and authorities in deciding whether to apply or license vaccines to reduce economic losses caused by AI.

## 1. Introduction

The term *influenza* originally referred to epidemics of acute, rapidly spreading catarrhal fevers in humans caused by viruses belonging to the *Influenzavirus A* genus of the *Orthomyxoviridae* family [1]. As an enveloped virus with a segmented RNA genome, it can readily alter its antigenic properties through mutation and genetic reassortment. The evolving nature of this virus necessitates the continuous development of new and effective vaccines to combat emerging strains. Additionally, its potential adaptation to new hosts heightens concerns about the risk of a zoonotic pandemic [2].

Orthomyxoviruses are recognized as the cause of significant numbers of natural infections and diseases, usually of the upper respiratory tract. Cases are commonly observed in humans, horses, dogs, domestic pigs, and various bird species. Sporadic infections have also been reported in mink and several marine mammals [3,4,5]. Recently, a fast-spreading epidemic caused by the highly pathogenic avian influenza (HPAI) A(H5N1) clade 2.3.4.4b virus was reported among dairy cattle in the United States [6].

This review aimed to identify all relevant publications on HPAI A(H5N1) clade 2.3.4.4b vaccines, especially the testing of their efficacy. For this analysis, the Boolean search term “Clade 2.3.4.4b AND vaccine” was used in the following databases: PubMed, PubAg, Scopus, Cochrane Library, and ScienceDirect. Only experiments conducted in an appropriate, verifiable, and reproducible manner were included, thereby increasing the reliability of the conclusions drawn. Inclusion criteria were clinical trials or randomized controlled trials evaluating vaccines against HPAI clade 2.3.4.4b in chickens, ducks, or geese. Papers were included if they measured viral shedding, hemagglutination inhibition (HI) titers, mortality, and morbidity. Additionally, studies had to include clade 2.3.4.4b as either the vaccine strain or the challenge virus. Multiple studies examined more than one vaccine; in such cases, only the results for clade 2.3.4.4b vaccines or vaccine groups tested against clade 2.3.4.4b were considered.

## 2. General Characteristics of Avian Influenza

Avian influenza (AI) is one of the most devastating poultry diseases, first identified in the 19th century. It is a viral disease caused by influenza type A viruses, affecting the respiratory, digestive, and nervous systems of multiple bird species, including domestic poultry and wild aquatic birds [7]. Infection in domestic poultry can result in various clinical outcomes (Table 1). The low pathogenic avian influenza (LPAI) biovariants generally cause asymptomatic infections or mild symptoms, such as respiratory disease and reduced egg production. At the same time, HPAI often leads to severe systemic disease with mortality rates reaching 100% [8]. It is commonly observed in gallinaceous birds [9] but has also been reported in domestic aquatic birds [10]. However, wild aquatic birds, such as ducks and geese, are more likely to serve as a reservoir for LPAI strains [9]. Wild waterfowl can shed the virus asymptomatically [5], challenging surveillance and eradication efforts. Since HPAI strains originate from LPAI strains [11,12], the spread of HPAI from LPAI-infected sources remains a significant concern. Moreover, influenza A viruses can undergo genetic changes through antigenic drift and antigenic shift. Antigenic drift occurs through point mutations in the viral genome, leading to minor changes in the virus. On the other hand, the antigenic shift involves genetic reassortment, resulting in significant viral changes that enable the virus to cross species barriers and circulate in new populations [13].

### 2.1. AI in Mammalian Species

Although AI was historically considered an avian disease and hosts of HPAI include a wide variety of birds [14], infections have been documented in various mammalian species, including cats, pigs, ferrets, mice, macaques, cattle, foxes, beavers, bears, seals, dogs, goats, tigers, and leopards, as well as humans [15,16,17,18,19,20]. The USDA has begun assessing the potential to develop an effective vaccine against the H5N1 bird flu virus in cattle. However, it has cautioned that it is too early to determine how long the process will take. The virus has infected multiple dairy herds across several U.S. states [21].

As a result of antigenic shift, avian-to-mammalian transmission can occur, with pigs serving as an intermediate host, as they possess both α2,6 and α2,3 receptors for influenza viruses, similar to birds [22]. Suppose a pig is simultaneously infected with a mammalian and an AI virus. In that case, gene reassortment may occur, potentially leading to the emergence of a novel virus capable of human transmission [23]. However, it is essential to note that pigs have demonstrated low susceptibility to H5N1 clade 2.3.4.4b [24]. While this mechanism suggests a potential role for pigs as mixing vessels in AI epidemiology, the question remains unsettled. Conflicting scientific opinions exist regarding the significance of pigs in this process, and further studies are needed to clarify their exact role [25].

Given that HPAI A(H5N1) has recently infected more mammalian species than ever before [19] and HPAI A(H5N1) clade 2.3.4.4b viruses have acquired a significant number of mammalian adaptation mutations in recent years [6,26,27] the risk of its turning into a human pathogen is growing. Reports of AI infections in humans through direct contact with infected animals have increased over the past decade [28], with some cases proving fatal [18]. One source of exposure is wet markets or live poultry markets, and the closure of these places has been associated with a greater prevalence of AI in humans [29,30]. Farmers and poultry workers are advised to take stringent measures to prevent infection [15,31,32,33]; however, in a minority of cases, humans have contracted HPAI without any known exposure to poultry [29].

### 2.2. Public Health Relevance

Certain HPAI strains are zoonotic and capable of causing infections in humans. It has been acknowledged that AI was likely the source of human influenza pandemics [34] in the past and has the potential to trigger future ones [35]. This is particularly true given that HPAI A(H5N1) has recently infected more mammalian species than ever [19]. Furthermore, HPAI A(H5N1) clade 2.3.4.4b viruses have acquired a significant number of mammalian adaptation mutations in recent years [6,26,27]. A potential route of AI transmission involves direct avian-to-human transmission, followed by subsequent human-to-human spread [23]. Most reports of cases of AI in humans have claimed the infected person was in contact with birds either in wet bird markets [36,37] or farms [15,38]. However, human-to-human transmission has also been reported [39] as limited and generally non-sustained. It has been observed in small clusters or groups [40] and is assumed to pose a low risk presently [31,41].

Clinical signs in humans include fever, respiratory symptoms, vomiting, diarrhea, and, in severe cases, death [32,33]. Children, in particular, are at a high risk of death from H5N1 [13]. Recommended protective measures include wearing appropriate protective clothing when handling poultry, implementing health monitoring systems for human populations, and vaccination. However, the current vaccine supply is insufficient to meet the probable demand during a pandemic. Prophylactic and therapeutic treatments, such as adamantanes and the newer class of neuraminidase (NA) inhibitors, are also available [13].

### 2.3. Transmission

In humans, influenza viruses can be transmitted through the air and direct contact with an infected host. Airborne transmission is facilitated by temperate weather with low humidity, which aligns with the general perception of “flu season” [42]. Although AI is an influenza virus, human-to-human airborne transmission has not been observed thus far. However, airborne transmission without an intermediate host has occurred in mammals such as ferrets [43] and cattle [6]. In poultry, a study by Zhao et al. [44] suggested that airborne transmission could seriously impact farm-to-farm transmission. However, the airborne transmission of HPAI clade 2.3.4.4b between farms was shown to be limited in experimental settings [45] and, more recently, in the field, where James et al. [46] reported that the virus travelled only short distances (10–80 m). The probability of airborne infection is thought to be influenced by the farm type, housing type, flock size, flock density, and the number of infected animals [47]. A decreased flock density and improved husbandry, biosecurity, and ventilation measures could help limit airborne transmission [44].

Far more effective than airborne transmission, contact transmission is thought to be responsible for most cases of AI [47]. Non-airborne transmission sources and fomites, such as contaminated feathers, spread to 80 m [46]. Therefore, wild birds and mammals pose a risk to farm birds as they can enter farms and spread the virus via direct contact with farm animals or contaminate food sources. Surveillance and prevention of wild animal exposure may be helpful to reduce this risk [48]. In addition, indoor farming can help to reduce the risk of contamination via the droppings of migratory birds, which sometimes flock in great numbers.

Transmission of AI from wild to domestic birds occurs more frequently than the reverse, but dispersal among similar groups (e.g., wild-to-wild or domestic-to-domestic) is most efficient [49].

## 3. Pathogenicity, Clinical Symptoms and Lesions in Poultry

AI is commonly classified into LPAI and HPAI [50]. LPAI is typically associated with localized infections in the respiratory and alimentary tracts and generally has a low mortality rate (<5%) [9]. Its primary impact on the poultry industry relates to flock welfare, decreasing egg and meat production [9,51,52]. In contrast, HPAI had a severe effect on the poultry industry [53,54,55], causing systemic infections and high mortality rates. It can lead to respiratory disease, hemorrhages, necrosis, and inflammation of multiple organs [9]. Some possible clinical signs and lesions are listed in Table 1.

Gallinaceous species are highly susceptible to HPAI. Severe clinical disease and death develop rapidly after infection. Mortality can reach 100% in affected large-scale holdings within a few days. Domestic waterfowl often exhibit a more protracted course of disease. The clinical outcome might be mild or even pass unnoticed, depending on the circulating virus strain. However, in some cases, infection can be devastating. The disease severity and mortality among wild birds can vary significantly [2,50,51]. The outcome of the infection depends on multiple factors, including viral adaptation to the host, host immunogenicity, affected bird species, and the specific strain involved [9]. AI strains are classified based on their hemagglutinin (HA) and neuraminidase (NA) surface proteins [50]. Since influenza type A viruses continuously mutate through antigenic drift [13], HPAI can emerge from LPAI through intrinsic mutations in poultry hosts [56]. Currently, HPAI mutations have only been observed in LPAI strains of H5 and H7 HA subtypes, as all known HPAI viruses belong to either the H5 or H7 subtypes, whereas LPAI strains can belong to any of the 18 HA subtypes [50]. The exact factors driving this seemingly spontaneous shift remain unknown [11,29]. Both H5 and H7 strains have been detected in chickens [57] and in humans [35,57], making them strains of concern for both the poultry industry [51,58] and public health [35].

## 4. Economic Significance of AI

AI is responsible for the deaths of millions of birds due to disease or culling [59], making it a significant economic burden for poultry farmers [53,54,60]. The virus is widespread globally, with outbreaks reported in Europe [61,62], Asia [63], North America [64,65], Africa [66,67], and Australia [68].

Over the past two decades, AI outbreaks have sharply increased, leading to devastating economic losses in the poultry industry. Between 2005 and 2020, outbreaks resulted in the deaths of 246 million birds worldwide [2,69]. Since January of 2022, more than 130 million birds have been affected in the United States alone [70], along with approximately 3500 outbreaks reported in Europe between 2020 and 2024 [71]. These outbreaks have incurred billions of USD in costs due to the direct impacts of infection and culling, trade restrictions, and the adverse effects on public perception [7].

Once a flock is proven to be infected with HPAI and the birds are therefore culled, nearby farms have several options, depending on the country or region. These options include enhanced biosecurity and establishing quarantine, culling the entire flock with financial compensation, or vaccinating flocks in the area [72]. In Germany, Australia, the Netherlands, Belgium, the United States, China, Ethiopia, Thailand, Senegal, Mauritania, and other countries, farmers may be compensated for up to 50% of the value of a healthy bird [54,72,73,74] with funds provided by the government or through government-industry partnerships [2,74]. However, while compensation helps mitigate economic losses, it may not be sufficient to incentivize farmers to cull their flocks [72]. In some developing countries where farmers receive no compensation, selling infected birds to live markets may be seen as a way to reduce financial losses. However, this practice poses serious risks for epidemic prevention.

The economic impact of AI on poultry farming increases the incentive to implement biosecurity and other preventive measures, which can reduce the spread of the disease [54,73,75].

AI causes instability in the poultry sector. High mortality rates, reduced production, and declining consumer demand for poultry products result in significant income losses, forcing some farmers to abandon poultry farming in search of alternative sources of income [53]. Since rural livelihoods and food security depend on the eradication of HPAI, controlling the disease is of paramount importance.

Additional costly factors include expenses related to disease prevention [76,77] and the cost of medical treatment in cases of human infection [78].

All these factors led to the introduction of vaccination in developing countries, and even countries practicing the “stamping out” policy are sometimes under pressure from the poultry industry to work out vaccination strategies for their poultry stocks in case of emergencies [59].

## 5. Vaccination of Domestic Poultry Against AI

Vaccine protocols for AI can be designed for control, prevention, emergency response, or routine use [72]. However, their use remains controversial and is not widely adopted [79]. Multiple factors influence the reluctance to vaccinate poultry against HPAI. One key reason is the reliance on non-vaccinated, highly susceptible poultry as “sentinel” animals to detect emerging HPAI strains, enabling early identification and control measures. This approach is considered suboptimal due to potential virus entry and management challenges. Ethical and practical concerns further complicate this issue, as relying on non-vaccinated poultry for surveillance could be regarded as unethical, mainly when effective vaccines are available to prevent suffering and reduce the need for mass culling. The “stamping out” protocol was implemented in many countries to prevent the spread of new variants [80].

Economic and trade considerations also significantly complicate the hesitation to vaccinate. Countries importing poultry products often restrict imports from regions that use HPAI vaccines. These restrictions arise from the difficulty of distinguishing between infected and vaccinated birds, complicating disease surveillance and control. Vaccine efficacy and logistical challenges also contribute to the reluctance. Vaccines may not provide complete protection against rapidly mutating HPAI strains, and vaccinating large poultry populations can be logistically complex and costly [72].

Emerging trends in vaccine technology are addressing some of these concerns. These advancements offer promising solutions by enabling improved disease control and monitoring. However, widespread implementation of these innovations will require overcoming the economic, logistical, and regulatory barriers influencing vaccination strategies. In most developed countries, the financial losses caused by culling and restrictions in case of an outbreak are tolerable compared to the risk of an emerging human pandemic, which may pop up due to HPAI strains circulating undetected in vaccinated stocks.

Of the 15 countries that used around 115 billion doses of vaccines between 2002 and 2012, five (China, Egypt, Indonesia, Vietnam, and Hong Kong) implemented vaccination as a control strategy, while the other 10 countries (listed in Figure 1) used vaccines for emergency or preventive purposes [59]. AI is highly prevalent in Egypt because the country is intersected by numerous migratory pathways of avian species [81]. Consequently, China, Indonesia, Vietnam, and Egypt are engaging in regular vaccination protocols to combat AI [72].

Since then, cumulative data have not been available on vaccination in the countries where it is allowed, but studies address the experiences of poultry veterinarians who apply for vaccination [82,83,84]. Based on this information, millions of vaccine doses might have been used.

Other countries have established AI antigen reservoirs for the rapid development of emergency vaccines during outbreaks [2,85,86], and 13 countries have been reported to have emergency vaccine banks [72]. Surveillance programs indicated a shift in the dominant circulating virus clade, with clade 2.2.1 becoming less prevalent and clade 2.3.4.4b emerging as the dominant strain [81]. Many commercial vaccines target previously dominant clades, such as 1, 2.2.1.1, 2.2.1.2, 2.3.2, and 2.3.4. Evaluating whether these commercial vaccines remain effective against the current circulating H5NX clade 2.3.4.4b strain is essential. Licensed and unlicensed vaccines are listed in Table 2. Among the licensed vaccines, 95.5% are inactivated vaccines, while 4.5% are live vector vaccines [59].

Although the United States Department of Agriculture (USDA) has not implemented vaccination strategies, it has licensed and stockpiled over 200 million vaccine doses following outbreaks of AI. The following vaccines are licensed by the USDA:Recombinant vectored vaccine using the herpesvirus of turkeys (HVT);Reverse-engineered vaccine of H5;RNA particle vaccine using alphavirus RPH5;H5 modified DNA/plasmid DNA vaccine.

Several companies have developed innovative vaccines. CEVA and Boehringer Ingelheim produced a recombinant vectored HVT vaccine. Zoetis created a reverse-engineered H5 vaccine. Merck (Harris) developed an RNA particle vaccine utilizing alphavirus RPH5, and Huvepharma produced the AI H5 modified DNA/plasmid DNA vaccine [87].

Evaluating vaccination strategies allows for comparisons of different approaches against AI, helping identify the most effective protocols based on the specific context of a country, production system, and available resources. This approach can also be tailored to determine the most suitable protocol for individual farms [88].

**Table 2 vaccines-13-00453-t002:** Licensed and unlicensed avian influenza vaccines, based on data from [89].

Vaccine Type	Vaccine Status	Additional Information
Inactivated AIV	licensed vaccine	licensed in multiple countries for use in chickens, geese, turkeys, ducks, and other bird species
rHVT live vectored	licensed vaccine	licensed in the USA and Egypt for use in chickens
rNDV	licensed vaccine	licensed in Mexico and China for use in chickens
rFPV	licensed vaccine	licensed in the USA, Mexico and China for use in chickens, geese and Muscovy ducks
RNA	licensed vaccine	licensed for use in ducks and geese and was used in France
Live AIV	unlicensed vaccine	live wild-type LPAI virus or attenuated LPAI
Live vectored	unlicensed vaccine	Replication of defective adenovirus, avian leukemia virus, duck enteritis virus, infectious laryngotracheitis virus, attenuated salmonella typhimurium, vaccinia, replication-defective, Venezuelan equine encephalitis virus
In vitro-produced hemagglutinin	unlicensed vaccine	Baculovirus in insect cell culture, eukaryotic systems, plant or cell culture
DNA	unlicensed vaccine	Naked DNA

AIV—avian influenza virus, LPAI—low pathogenic avian influenza, rFPV—recombinant fowl pox virus, rHVT—recombinant herpesvirus of turkey, rNDV—recombinant Newcastle disease virus.

## 6. Clade 2.3.4.4b

AI viruses of HPAI H5N1 clade 2.3.4.4 have become dominant worldwide since 2014 [90], with outbreaks of subclade 2.3.4.4b HPAI emerging in Europe around 2020 [91]. Currently, this clade remains the most dominant worldwide.

HPAI caused the deaths of over 23 million domesticated birds in 2020, more than 61 million in 2021, over 140 million in 2022, more than 77 million in 2023, and over 78 million in 2024 worldwide [92,93]. HPAI H5N1 was responsible for 400 outbreaks globally in 2020, which increased to 3032 in 2021 and 2401 in 2022 before decreasing to 505 in 2023 and remaining relatively stable with 523 outbreaks in 2024 [94]. However, clade-specific data are lacking.

Additionally, it raises concerns regarding potential epizootic outbreaks [90]. Subclade 2.3.4.4b has been detected in carnivores [95,96] and reported in humans [89]. Due to the significant threat posed by subclade 2.3.4.4b to the poultry industry and public health, this review focuses on strategies to mitigate its impact on poultry production through vaccination. While public health considerations often emphasize the sentinel role of non-immunized domestic poultry and the implementation of a “stamping out” policy, this review prioritizes the evaluation of vaccination as a control measure.

The review first explores current data on the efficacy of both licensed and unlicensed vaccines against subclade 2.3.4.4b. It assesses the protective efficacy (PE) of these vaccines by comparing morbidity and mortality rates, viral shedding, and HI titers in vaccinated versus unvaccinated populations. Additionally, it examines the practical and immunological factors influencing vaccine PE and the effectiveness of different vaccination programs.

## 7. Evaluation of Published Data

The Boolean search yielded 25 results from PubMed, with eight publications meeting the inclusion criteria. PubAg provided nine papers, of which only one was eligible. The Cochrane Library returned no relevant research papers. Scopus produced 28 results, but none met the predefined selection criteria and were excluded from this review. Lastly, ScienceDirect yielded 45 results, with four eligible papers.

Thirteen eligible papers were identified. The results are summarized in the PRISMA diagram in Figure 2. The examined studies investigated various vaccine types listed in Table 3.

### 7.1. Studies Used in Summarizing the Latest Research

This review encompasses the last five years of publications from 2019–2023. Most studies were limited to 10 chickens per vaccine group, with the largest group size reported by [98], which included 30 chickens. The trials examined vaccinated and unvaccinated chickens and evaluated the results following the challenge (infection with the field virus).

Many vaccine trials involving chickens were conducted on SPF White Leghorns. However, two studies utilized commercial breeds: Ross 308 and Tetra-SL [99] and Avian 48 [100]. The age at vaccination varied across studies, ranging from one day to six weeks of age.

Of the three studies that involved ducks, one focused on Muscovy and Pekin ducks [101], another on the hybrid of these two breeds known as the mule duck [102], and the third on a local Chinese breed, the Shaoxing shelduck. The same study that examined vaccines in geese focused on a Chinese breed, the Shitou goose [103]. The trials are summarized in Table 3, while the results are summarized in Table 4 and Table 5.

**Table 3 vaccines-13-00453-t003:** The vaccine types, their advantages and disadvantages, and the number of trials focusing on each vaccine type.

Vaccine Type	Number of Trials Focused on Vaccine Type	Advantages	Disadvantages	References
Inactivated vaccine	6	Safe as the pathogen is killed, stable, strong humoral immunity	Adjuvants are required to enhance immune response, requires multiple doses or boosters for strong immunity, induces weaker cell-mediated immunity	[100,104,105,106]
Inactivated reverse genetics vaccine	4	Allows precise modifications for improved safety and immunogenicity, strong humoral immunity	Requires specialized laboratory techniques and equipment, limited cellular immunity, expensive and complex to manufacture, may still require boosters to maintain immunity	[85,86,98,107]
Recombinant vectored vaccine	6	Safe, can induce both humoral and cellular immunity, no interference with maternal antibodies, potential for multivalent protection against more avian diseases, potential for rapid adaptation to emerging H5N1 clade variations	Requires a suitable viral vector, which may cause pre-existing immunity issues, expensive and complex to manufacture, virus shedding is still possible, requires specific storage	[99,100,102,108]
Recombinant subunit vaccine	2	Safe, low risk of adverse reaction, allows precise modifications	Requires adjuvants and multiple doses to achieve strong immunity, less likely to induce strong cellular immunity expensive and complex to manufacture, requires specific storage	[101,102]
Virus-Like Particle vaccine	1	Safe, strong immune response, can induce both humoral and cellular immunity, allows precise modifications	Requires an adjuvant or booster for long-term immunity, expensive and complex to manufacture, long development time, requires specific storage conditions	[102]

**Table 4 vaccines-13-00453-t004:** Summary of vaccine trials and key parameters.

Vaccine Type and Strains	Challenge Virus and Dose	Antigenic Relativeness of Vaccine Strain to Challenge Virus	Species and Age of Vaccination and Method	Age at Challenge and Method	Reference
Trivalent inactivated (clade 2.2.1.1 and 2.2.1.2.)	2.2.1, 2.2.1.1, 2.2.1.2, 2.3.4.4b H5N8, 10^6^ EID_50_/0.1 mL	89.9%	chicken, 2 weeks old, subcutaneous	4 wpv, intranasal	[104]
Comparison of inactivated bivalent H5 + ND7 (2.3.4.4b) to H5 plus NDVg7 (clade 2.2.1.2, 2.3.4.4b)	H5N8, 10^6^ EID_50_/0.1 mL	not evaluated	chicken, 2 weeks old, subcutaneous	3 weeks old, intranasal	[105]
vHVT vaccine expressing COBRA of AI or vHVT-ND-AI or vHVT-IBD-AI	H5N2 2.3.4.4A, H5N8 2.3.4.4B, H5N1 2.2., 10^6^ EID_50_/0.1 mL	not evaluated	chicken, one day old subcutaneous	4 wpv Avian influenza: intra-choanal, IBDV: eye drop route, NDV: intramuscular route	[108]
rHVT-H5 (clade 2.2) alone or with inactivated rgH5N1 (clade 2.2.1.1 and clade 2.2.1.2)	H5N8 2.3.4.4b, 10^6.3^ EID_50_/0.1 mL	not evaluated	chicken, one day old, rHVT-H5: subcutaneous, inactivated rgH5N1: intramuscular	One group, 28 days old, others 35 days old, intranasal and eye drop	[100]
Inactivated AI H5N8 strain vaccine vs bivalent inactivated AI vaccine with H5N1	H5N1 clade 2.2.1.2, H5N8 clade 2.3.4.4b, 10^6^ EID_50_/0.1 mL	92.9–100%	chicken, 2 weeks old, subcutaneous	3 wpv, intranasal	[106]
Inactivated bivalent reverse genetics vaccine clade 2.3.2.1d and 2.3.4.4b	H5NX 2.3.4.4b, 10^6^ EID_50_/0.1 mL	2.5 antigenic units distance	chicken, 5 weeks old, unspecified	not specified, unspecified route	[86]
Inactivated reverse genetic AI vaccines H5N6 clade 2.3.4.4b	H5N6 clade 2.3.4.4b, 10^6^ EID_50_/0.1 mL	homologous	chicken, 6 weeks old, intramuscular	9 weeks old (3 wpv), intranasal	[85]
Reverse genetics inactivated avian influenza H5N1 vaccine clade 2.3.4.4	H5N8 clade 2.3.4.4b, 10^6^ EID_50_/0.1 mL	96.1%	chicken, 21 days old for first vaccine and 45 days old at booster (prime-boost), subcutaneous	25 days after the booster oculonasal route	[107]
Comparison of three vaccines 1, clade 2.3.4.4b H5 virus-like particle (some received a booster), 2, Clade 2.3.2 H5 subunit vaccine, 3, clade 2.3.4.4b H5 RNA particle vaccine	H5N8 clade 2.3.4.4b, 4.6 × 10^5^ EID_50_/0.1 mL	Not evaluated	mule duck, 3 weeks old (or 1 day old if from prime and booster group), subcutaneous	6 weeks old (3 wpv), intraocular	[102]
rHVT-H5 (clade 2.2)	H5N8 clade 2.3.4.4b, 10^6^ EID_50_/0.2 mL	92%	chicken, 1 day old, subcutaneous	5 weeks old, oronasal route	[99]
Clade 2.3.2 H5-recombinant baculovirus vaccine	H5N8 clade 2.3.4.4b, 10^6^ EID_50_/0.1 mL	Not evaluated	Muscovy/Pekin duck, 10 days old (and on 28 days old if booster), subcutaneous	31 days old (if single dose) 49 days old (if double dose), intranasal route	[101]
Trivalent inactivated H5N6, H5N8 clade 2.3.4.4b, H7N9	H5N1, H5N6, H5N8 clade 2.3.4.4b, 10^6^ EID_50_ or 100 DLD_50_	similar	chicken, Shaoxing Shelduck, Shitou goose, chicken: 3 weeks old, duck and goose: 2 weeks old, intramuscular	3 wpv, intramuscular	[103]
H5/H7 trivalent reverse genetics inactive vaccine strains H5N6 clade 2.3.4.4b (with H5N1 clade 2.3.2.1d and H7N9)	H5N1, H5N6, H5N8 clade 2.3.4.4b, 10^5^ EID_50_	Not stated	chicken, 3 weeks old, intramuscular	3 wpv, intramuscular	[98]

AI—avian influenza, DLD_50_—50% duck lethal dose, EID_50_—egg infective dose 50, IBD—infectious bursal disease, ND—Newcastle disease, vHVT—vectorized herpesvirus of turkeys, wpv—weeks post-vaccination.

**Table 5 vaccines-13-00453-t005:** Summary of vaccine trial results.

Vaccine Type and Strain Used	HI Titer (log2)	Mortality	Clinical Signs	Shedding	Reference
Trivalent inactivated (clade 2.2.1.1 and 2.2.1.2.)	>6 (at 2 weeks)	0%	93.3% protected	shedding in 20% of vaccinated birds at 3 dpc, decreased to ~7% at 6 dpc, 1.7–2.1 log10 EID50	[104]
Comparison of inactivated bivalent H5 + ND7 (2.3.4.4b) to H5 plus NDVg7: clade 2.2.1.2, 2.3.4.4b	<3	ValleyVac H5plus NDVg7: 80% protection, H5 + ND7: 20% protection	ValleyVac H5plus NDVg7: 80%, H5 + ND7: 20% protection	decreased virus shedding, H5Plus NDVg7: 1.31–2.70 log10 EID50, H5 + ND7: 3.84–6.52 log10 EID50	[105]
vHVT vaccine expressing COBRA of AI or vHVT-ND-AI or vHVT-IBD-AI	>6	all survived except 1 euthanized and one vHVT-ND-AI vaccinated; Egypt/14 challenged died	all protected except one vHVT-ND-AI vaccinated, Tk/Hungary/16 challenged birds	decreased significantly, 2–6 log10 EID50	[108]
rHVT-H5 (clade 2.2) alone or with inactivated rgH5N1(clade 2.2.1.1 and clade 2.2.1.2)	>6 in both vaccinated groups	50–60% if challenged at 28 days old, if challenged at 35 days old, prime-boost: 0% and rHVT-H5 alone: 20%	milder than non-vaccinated groups but not fully protected	decreased significantly, 0.35–4.95 log10, but lasted 5 days	[100]
Inactivated AI H5N8 strain vaccine vs bivalent inactivated AI vaccine with H5N1	>6 in week 2 in both groups	0%	0% in vaccinated groups	decreased significantly, 2.63–3.43 log10 no viral shedding after 7 dpc	[106]
Inactivated bivalent reverse genetics vaccine clade 2.3.2.1d and 2.3.4.4b	>8 at full dose	all birds with full dose survived, 5 died at 1/100th dose	0% in the group challenged with 2.3.4.4b	no viral shedding at full dose	[86]
Inactivated reverse genetic AI vaccine H5N6 clade 2.3.4.4b	>8 at full dose	0% at full and 1/10th dose	0% at full and 1/10th dose	no viral shedding at full or 1/10th dose	[85]
Reverse genetics inactivated avian influenza H5N1 vaccine clade 2.3.4.4	not evaluated in log form, GM: 1713, ±1GSD: 800–4003	0%	0%	20% shed at low levels first 3 dpc, 2.73–3.08 log10 EID50	[107]
Comparison of three vaccines 1, clade 2.3.4.4b H5 virus-like particle (some received a booster), 2, Clade 2.3.2 H5 subunit vaccine, 3, clade 2.3.4.4b H5 RNA particle vaccine	vaccine 1: 4, vaccine 1 + booster: 9, vaccine 2: not positive, vaccine 3: 3	vaccine 1: 1 died, vaccine 3: 1 died	vaccine 1: 7 showed CS, vaccine 1 boosted: 3/9, vaccine 2: 4/10, vaccine 3: 3/10	vaccine 1: 3 shed up to 3 dpc, vaccine 1 boosted: 10% to day 3, vaccine 2: all shedding up to 15 days, vaccine 3: 3 ducks up to day 7, 1–6 log10	[102]
rHVT-H5 (clade 2.2)	range of 2.5–7	0%	0–10% showed CS	7/20 birds in direct challenge shed 2–4 log10 EID50, no shed in the contact group	[99]
Clade 2.3.2 H5-recombinant baculovirus vaccine	>4 in Muscovy, >4 in Pekin	10% Muscovy duck (single dose), 0% Muscovy in double dose, 0% Pekin in both single and double dose	100% Muscovy: depression (single dose), 0% Pekin	Muscovy: no shedding day 7, Pekin: 0%	[101]
Trivalent inactivated H5N6, H5N8 clade 2.3.4.4b, H7N9	>8 in chicken, >6 in duck, >4 in goose	100% protection	100% protection	0%	[103]
H5/H7 trivalent reverse genetics inactive vaccine strains H5N6 clade 2.3.4.4b (with H5N1 clade 2.3.2.1d and H7N9)	2.4 in bivalent, 7.6 in trivalent	bivalent: 40–50% died except in H7N9 challenged group, trivalent: complete protection	bivalent: not stated, trivalent: complete protection	bivalent: shed 2–5log10 EID50 titers of viruses through both the oropharynx and cloaca, trivalent: no shedding	[98]

AI—avian influenza, CS—clinical signs, dpc—days post-challenge, GM—geometric mean, GSD—geometric standard deviation, IBD—infectious bursal disease, ND—Newcastle disease, rHVT—recombinant herpes virus of turkey, vHVT—vectorized herpesvirus of turkeys.

### 7.2. Clinical Signs

Clinical signs were monitored in all studies and are listed in Table 1, with recorded results presented in Table 5. The presence or absence of clinical symptoms was documented to assess vaccine efficacy. The majority of vaccines provided high levels of protection against clinical signs of the disease in vaccinated groups. However, it should be noted that not all vaccines offer complete or uniform protection. Variations in the PE of different vaccines ranged from complete protection to mild symptoms in some or many birds [85,86,99,103,104,106,108]. Nevertheless, vaccination greatly reduced the severity of clinical signs and the number of affected birds.

Azab et al. [105] compared two vaccines: an AI and Newcastle disease (ND) vaccine. They found that when challenged with H5N8, 20% of birds exhibited clinical signs if vaccinated with the Valley Vac H5Plus NDVg7 2.2.1.2, 2.3.4.4b vaccine, whereas 80% showed clinical signs when vaccinated with the inactivated AI and ND vaccine, H5 + ND7 (2.3.4.4b).

El-Shall et al. [100] investigated a turkey herpes virus vaccine administered alone or combined with an inactivated H5N1 vaccine. Their findings indicated that these vaccines did not protect day-old chickens against morbidity.

Niqueux et al. [102] found that mule ducks were not fully protected from clinical signs by any of their three HA-based vaccines tested. One of the vaccines was an adjuvanted H5N1 subtype viral pseudo-particle vaccine, incorporating a modified H5 hemagglutinin from the clade 2.3.4.4b HPAI virus and an N1 neuraminidase from the clade 2.1.3.2 HPAI virus. The second vaccine was a commercial bivalent adjuvanted vaccine containing a modified H5 from the clade 2.3.2 HPAI virus and an inactivated avian orthoavulavirus type 1 LaSota strain. The third vaccine was a viral pseudo-particle vaccine consisting of a recombinant, non-productive subgenomic RNA replicon derived from the Venezuelan equine encephalitis virus strain, expressing the H5 hemagglutinin of the clade 2.3.4.4b HPAI virus. However, the administration of a booster dose of clade 2.3.4.4b H5 subtype viral pseudo-particles vaccine decreased the prevalence of clinical signs compared to the single dose.

Sultan et al. [101] reported that 100% of Muscovy ducks vaccinated with a clade 2.3.3 H5-recombinant baculovirus vaccine have shown clinical signs, while Pekin ducks showed no clinical signs. In one study, researchers did not record any clinical signs [98].

### 7.3. Mortality

Mortality was recorded in the studies to assess vaccine efficacy, as seen in Table 5. All viruses used in the challenge tests caused high mortality in unvaccinated chickens. In contrast, several vaccine trials reported 0% mortality in vaccinated groups. These trials included three trivalent inactivated vaccines [98,103,104] a mono- and bivalent inactivated vaccine [106], three reverse genetics inactivated vaccines [85,86,107], and two rHVT-H5 vaccines—one of which achieved 0% mortality only with a prime-boost vaccination strategy [99,100]. Additionally, an H5-recombinant baculovirus vaccine demonstrated no mortality in Pekin ducks with both single and double doses, but in Muscovy ducks, only the double dose achieved 0% mortality. The single dose of the vaccine resulted in 10% mortality in Muscovy ducks. These findings suggested that the effectiveness of vaccination is species-dependent and can vary considerably [101].

Two studies reported only one or two deaths in the vaccinated groups [102,108]. These two studies included H5 virus-like particle, H5 subunit, and H5 RNA particle vaccines, a vHVT vaccine expressing COBRA of AI, and vHVT-ND-AI or vHVT-IBD-AI vaccines.

Moreover, Azab et al. [105] reported a 20% mortality rate in chickens vaccinated with the Valley Vac H5Plus NDVg7 (clades 2.2.1.2 and 2.3.4.4b) and an 80% mortality rate in those vaccinated with the inactivated AI and ND H5 + ND7 (clade 2.3.4.4b).

The age at which vaccines are administered and the number of doses given appear to influence their PE. El-Shall et al. [100] found that challenging chickens with an H5N8 clade 2.3.4.4b virus at an older age, followed by a booster vaccine, reduced mortality rates to 0%, compared to 50–60% mortality in younger chicks that did not receive a booster. In chickens challenged at an older age without receiving a booster vaccine, a mortality rate of 20% was observed.

### 7.4. HI Titer and Virus Shedding

HI, titer and virus shedding results can be seen in Table 5. The cut-off values of HI titers differ depending on the amount of virus applied in the tests. HI, titers greater than 3 log₂ were considered positive in certain studies [101,102] at a dilution of ≥4 log_2_ against 4 hemagglutinating units (HAU) of antigen, according to WOAH standards. However, some laboratories use 8 HAU in HI assays, which is an acceptable modification, but this affects the cut-off level, and a positive titer is defined as ≥3 log_2_ [56].

In the reviewed studies, viral shedding through the oral or fecal route was assessed using tracheal and/or cloacal swab samples collected after challenging the vaccinated birds with AI virus. Several studies reported higher viral loads in tracheal swabs than in cloacal swabs [99,100,102,107].

Seven of the selected studies reported HI titers above 5 log_2_ in all of their trials, which correlated with a reduction in mortality and morbidity and decreased shedding [85,86,100,103,104,106,108].

Trivalent reverse genetics inactivated vaccines with HI titers greater than 5 log_2_ showed no virus shedding in all poultry species studied [103]. In other studies, a trivalent inactivated vaccine (clades 2.2.1.1 and 2.2.1.2) with HI titers higher than 5 log_2_ initially showed virus shedding in 20% of birds, which decreased to 7% after six days. Virus shedding ranged between 1.7–2.1 log_10_ EID_50_ [104].

No shedding was observed in Pekin and Muscovy ducks vaccinated with a clade 2.3.2 H5-recombinant baculovirus vaccine [101]. Similarly, an inactivated reverse genetics AI vaccine (H5N6 clade 2.3.4.4b) with high HI titers demonstrated no virus shedding [85]. A bivalent reverse genetics inactivated vaccine with HI titers lower than 3 log_2_ resulted in oropharyngeal and cloacal virus shedding levels between 2–5 log_10_ EID_50_ [98]. However, chickens vaccinated with a similar vaccine and the same clades showed no shedding when administered the full vaccine dose [86]. Birds vaccinated with another reverse genetics inactivated AI H5N1 vaccine (clade 2.3.4.4) 20% shed the virus, with virus titers ranging from 2.73–3.08 log_10_ EID_50_ [107].

Monovalent and bivalent inactivated H5N8 and H5N8/H5N1 AI vaccines with high HI titers showed low virus shedding levels for up to 3 days, disappearing by day 7 [106].

The inactivated H5Plus NDVg7 vaccine resulted in lower virus shedding, titers 1.31–2.70 log_10_ EID_50_, compared to the H5 + ND7 vaccine, 3.84–6.52 log_10_ EID_50_. Both were associated with HI titers lower than 3 log_2_ [105].

The rHVT-H5 vaccine (clade 2.2), whether used alone or with an inactivated rgH5N1 vaccine (clades 2.2.1.1 and 2.2.1.2), showed virus shedding in vaccinated birds. However, booster administration reduced shedding levels to <1 log_10_ [100]. In another study, birds vaccinated with the rHVT-H5 vaccine (clade 2.2) and showing HI titers of 2.5–7 log₂ shed the virus in 35% of directly challenged birds, but no shedding was observed in contact birds [99].

Vaccines such as vHVT-AI, vHVT-ND-AI, or vHVT-IBD-AI resulted in shedding in 1–9 birds out of 10 in each group, with titers ranging from 2–6 log_10_ EID_50_, despite high HI titers above 6 log₂ [108].

H5 virus-like particle, H5 subunit, and H5 RNA particle vaccines (clade 2.3.4.4b) demonstrated varying levels of shedding, with 10% to all birds shedding virus at levels between 1–6 log_10_ EID_50_. Even birds given a booster vaccine occasionally shed the virus, although higher HI titers were associated with lower levels of shedding [102].

According to scientists, a higher HI titer generally correlates with lower mortality and reduced virus shedding [109,110]. However, it is important to note that the HI assay measures only antibody binding to HA receptors; it does not account for the presence of other proteins that may enhance the immunogenicity of HA and NA [86].

The findings summarized in this review partly supported this statement. However, exceptions and variations were present in the data.

### 7.5. Maternally Derived Antibodies (ABs)

Although many of the studies we reviewed were conducted on chicks lacking maternal ABs against AI viruses, several vectored vaccine trials assessed the presence of maternal ABs prior to vaccination [99,101,102]. El-Shall et al. [100] studied commercial chicks with maternal ABs and found their presence influenced vaccine efficacy. Specifically, maternal ABs were correlated with reduced body weight and increased mortality in chickens challenged with AI at four weeks of age compared to those challenged at five weeks. In contrast, Palya et al. [99] investigated only commercial chickens (broilers and layers) without maternal ABs.

Due to the short production cycle of broilers, PE must be achieved earlier than five weeks of age; otherwise, vaccination may not be cost-effective. Prime-boost vaccination strategies may mitigate the impact of maternal ABs and enhance vaccine PE [100]. However, such a strategy would necessitate additional labor for poultry farmers, potentially reducing on-farm compliance.

Live vectored vaccines are typically administered in ovo or at one day of age. However, these vaccines are ineffective if ABs—whether maternally derived or from a previously administered vaccine—against the viral vector are present in the animal [109]. This issue may arise in chicks previously vaccinated for fowlpox or Marek’s disease using vectored vaccines [2]. Given the impact of ABs on PE, routine screening for ABs against vectored vaccines should be standard practice in trials evaluating vectored vaccines.

### 7.6. Antigenically Distant Field Viruses and Vaccine Strains

Many commercial vaccines are available for AI. However, the genetic variability of the currently circulating AI strains poses a significant challenge to the efficacy of vaccination strategies. Genetic and antigenic compatibility are critical concerns, as failure to protect against field strains may accelerate antigenic drift and promote new mutations, further exacerbating the spread of HPAI [106].

A review by Mo et al. [110] suggested a correlation between vaccine-induced protection—characterized by a reduction in clinical signs, viral shedding, and mortality—and the genetic compatibility between the circulating virus and the vaccine seed strain. However, vaccination with antigenically distant recombinant viruses, such as the herpesvirus-vectored H5 vaccine (rHVT-H5, clade 2.2), has been shown to provide cross-protection against viruses from the H5NX clade 2.3.4.4b [100].

#### 7.6.1. Single Dose vs. Multiple Dose

Typically, a single dose of AI vaccine is administered in these experiments. While many studies have demonstrated the effectiveness of a single dose, the duration of long-term protection has not been thoroughly evaluated [85,106]. Swayne et al. [2] suggested that a single dose of the AI vaccine may be sufficient for broilers due to their shorter production cycle. However, chickens and ducks may require two or three doses under field conditions to achieve adequate protection. This is particularly relevant for ducks, which tend to exhibit a weaker immunogenic response compared to chickens. Moreover, the immune response in ducks can vary significantly between breeds, necessitating tailored vaccination strategies to ensure adequate disease control [101,102]. Maartens et al. [107] reported that serological and humoral responses against the viral antigen increased with the administration of additional booster vaccines. Furthermore, in one study, clinical signs and mortality rates decreased significantly following an additional booster dose [100].

#### 7.6.2. Monovalent vs. Multivalent

No vaccine is effective against all 16 HA subtypes of AI [2]. However, co-circulating multiple strains during an outbreak poses a significant challenge to disease control. In such scenarios, multivalent vaccines, including bivalent and trivalent formulations, offer the advantage of targeting multiple strains simultaneously, potentially improving overall efficacy [86,105]. Additionally, bivalent vaccines can provide practical benefits, including increased convenience and cost-effectiveness for farmers [86,106].

A study by Ibrahim et al. [106] found that a bivalent vaccine significantly reduced viral shedding compared to a monovalent vaccine. However, the monovalent vaccine induced higher HI AB titers, suggesting potential antigen interference when using multivalent vaccines. This refers to antigenic competition, where interactions between antigens may reduce the efficacy of vaccines containing multiple epitopes. Such interference may affect the seroconversion of one strain over another, as noted by the authors. Nonetheless, other studies have reported high HI titers with bivalent vaccines, indicating that antigen interference is not an inherent issue in all cases [86,103,104]. However, if it does persist, a multivalent prime-boost vaccination strategy might be beneficial [111]. A more comprehensive evaluation of the mechanisms behind antigen interference in multivalent vaccines is warranted to optimize their design and efficacy.

Trivalent vaccines have also shown promise in addressing the complexity of co-circulating strains. A trivalent inactivated vaccine containing H5N6 clade 2.3.4.4b, H5N1 clade 2.3.2.1d, and H7N9 strains was developed using reverse genetics. This molecular technique enables the creation of specific vaccine strains with reduced pathogenicity but robust immunogenic properties [98,112]. This vaccine demonstrated PE against all three challenge viruses, highlighting its potential for controlling outbreaks involving multiple strains [98]. Despite the complexities associated with developing and implementing multivalent vaccines, they represent a significant advancement in AI control. Further research should explore the trade-offs between broader protection and the risk of antigenic interference to ensure their optimal application in outbreak scenarios.

#### 7.6.3. Vaccine Type

Inactivated vaccines account for 95.5% of all AI vaccines used in the field worldwide, while live recombinant virus vaccines—primarily recombinant ND-vectored vaccines with AI H5 gene inserts—comprise only 4.5% [85]. However, inactivated virus vaccines are less immunogenic and may require multiple doses, making them more labor-intensive and potentially expensive [113]. Additionally, inactivated vaccines primarily induce humoral immunity, offering limited mucosal or cellular immune responses, which may affect their efficacy in preventing infection [114]. Nonetheless, due to the short lifespan of broiler chickens, the relatively short duration of protection provided by inactivated vaccines may not be a significant concern. In contrast, limited protection duration could present challenges for poultry with longer lifespans.

In a recent study, Maartens et al. [107] evaluated the efficacy of an inactivated H5N1 vaccine against a South African HPAI strain (clade 2.3.4.4b) in chickens. The vaccine, developed using reverse genetics, provided complete protection against clinical signs and mortality. The study demonstrated that antigenically compatible inactivated vaccines can confer strong protection and significantly reduce viral shedding [107]. Despite their efficacy, inactivated virus vaccines have notable limitations, particularly regarding the sale of vaccinated birds, as discussed further below.

Given these constraints, the advantages of vector vaccines should be considered. First, vectored vaccines are recommended for administration either in ovo or on the day of hatching, which presents a logistical advantage [113]. Second, many vaccines utilize viral vectors to deliver AI antigens. HVT and baculovirus have been evaluated as vaccine vectors in experimental trials [101,108]. HVT-vectored vaccines have been shown to provide long-term immunity and can be combined with other vaccines [99]. Despite these advantages, vectored vaccines also have limitations. As previously mentioned, live vector vaccines are ineffective if ABs against the vector virus are already present in the bird before vaccination [2]. Another challenge is host susceptibility. Vectored vaccines may not be suitable for all poultry species depending on their ability to support replication of the vector virus [113]. Furthermore, WOAH does not recommend using live vaccines, as they could promote genetic reassortment and generation of HPAI viruses [56].

The third vaccine category identified in the reviewed studies includes subunit and nucleic acid vaccines. In one trial, researchers investigated the effects of virus-like particles and RNA particle vaccines in mule ducks, reporting a slight reduction in mortality, morbidity, and viral shedding [102]; the results can be found in Table 5. While these findings are promising, the high production cost of subunit and nucleic acid vaccines may limit their practical application [113]. This area warrants further research to assess their feasibility for large-scale implementation.

### 7.7. Poultry Species Considerations

Although ducks and geese often are asymptomatic carriers and shedders of AI, they can still exhibit clinical signs of infection [9]. It has been suggested that immunogenicity varies among poultry species [2]. For example, Zeng et al. [103] observed differences in HI titers for the same vaccine and the same vaccine and protocol across different poultry species, with chickens exhibiting the highest HI titers compared to ducks and geese. Furthermore, immunogenicity appears to vary even within different breeds of the same species.

One study found that Muscovy ducks were more sensitive to AI strains of differing pathogenicity within the same clade than Pekin ducks. Pekin ducks exhibited higher HI titers, lower levels of viral shedding, fewer clinical signs, and lower mortality rates than Muscovy ducks [101]. These findings align with those of Pantin-Jackwood and Swayne [9], who reported that Muscovy ducks are more susceptible to AI than many other duck species and concluded that disease manifestation varies depending on both duck breed and viral strain.

In trials involving chickens, Leghorn breeds have demonstrated stronger AB responses compared to ducks [2]. Ducks generally exhibit a weaker immune response than chickens, with considerable variability among duck breeds. Consequently, administering a double-dose vaccination regimen is often recommended for ducks [101,102]. Some researchers suggest that post-vaccination protection against HPAI in ducks may depend on mechanisms beyond humoral immunity alone [102].

Turkeys are reported to be more sensitive to HPAI, exhibiting higher mortality and morbidity rates than chickens. Transcriptome analyses have shown that turkeys respond differently to HPAI virus infections, displaying distinct gene expression patterns, particularly in RNA metabolism and immune response pathways. In turkeys, high virulence is associated with the activation of genes involved in transcription and translation rather than immune-related genes, along with widespread suppression of host gene expression related to protein synthesis and immune function. This suggests that turkeys experience more extensive cellular damage and immune system dysfunction during infection, leading to multiple organ failure. In contrast, chickens exhibit a more robust immune response, which helps limit viral spread and allows them to tolerate tissue damage to some extent [115].

This variability underscores the critical need for developing breed-specific vaccination strategies to achieve effective disease control.

### 7.8. Farm Conditions vs. Lab Conditions

Conventional methods for evaluating vaccine PE do not accurately simulate natural virus transmission, herd immunity, or transmission dynamics. Notably, none of the vaccination trials examined these factors. Viral and bacterial co-infections may also influence vaccine PE. However, this aspect was not assessed in most trials. An exception is the study by Palya et al. [99], which reported 40% mortality and 45% morbidity among unvaccinated birds in direct contact with infected, vaccinated birds. Many trials were conducted in negative-pressure biosafety level 3 (BSL-3) isolators. However, the strict biosecurity measures applied in vaccine trials cannot be fully replicated under commercial farm conditions, which may impact vaccine performance in real-world settings.

### 7.9. Method of Vaccination

Hatchery vaccinations are more effective in terms of cost, time, and protection [88]. These vaccinations are administered either in ovo or on the first day after hatching [116]. However, none of the studies reviewed specifically examined trials in which vaccines were administered to chicks less than one day old. This underscores the need for further research in this critical age group to better understand vaccine efficacy, immune responses, safety, and potential challenges associated with early vaccination.

As mentioned earlier, evaluating vaccination strategies is essential for developing the most appropriate protocols for AI vaccines tailored to the specific needs of individual farms [88].

#### 7.9.1. Differentiating Infected from Vaccinated Animals (DIVA)

The acronym DIVA stands for differentiating infected from vaccinated animals [117]. The presence of AI ABs is routinely used to test for AI infection. However, AI ABs can also indicate prior vaccination or previous infection, making it impossible to determine the source of immunity solely based on serological testing. The inability to distinguish healthy, vaccinated birds from infected, non-vaccinated birds is often cited as an argument against vaccination (Table 6), as many countries will not import live birds or poultry products in the presence of AI ABs, thereby impacting trade [117].

However, the DIVA strategy enables differentiation between vaccinated and infected poultry. When vaccines are developed, either additional antigens not present in the native virus can be included (positive marker vaccines) or specific antigens present in the native virus can be removed (negative marker vaccines) [113,117]. To implement the DIVA strategy, live vectored recombinant vaccines—which incorporate only parts of the virus—must be used, rather than inactivated virus vaccines, which contain the entire virus [2,113]. DIVA-compatible vaccines allow for the detection of wild-type virus-specific ABs using diagnostic tests, making it possible to differentiate animals vaccinated with a marker vaccine from those infected with the wild-type virus [117]. To facilitate both vaccination and disease surveillance, future AI vaccination trials should prioritize the implementation of DIVA protocols and the development of marker vaccines.

**Table 6 vaccines-13-00453-t006:** Reported reasons for implementing or not implementing vaccines in 69 surveyed countries [72].

Reasons to Vaccinate	Reasons for Not Vaccinating
Has been effective in eradicating HPAI outbreaks	Other methods have been successful in eradication
Decrease in morbidity	Risk of silent infections and shedders
Decrease in mortality	Difficult to distinguish infected from non-infected (without DIVA protocol)
Decrease in viral shedding	Vaccine costs
If the flock is at high risk (emergency)	Trade restrictions
-	Encourages leniency in biosecurity

#### 7.9.2. Incentives for Farmers

The most common AI control strategies focus on virus eradication by culling affected and high-risk bird populations. Vaccination is not widely implemented globally, with only a handful of countries routinely vaccinating against LPAI and even fewer countries against HPAI [59]. In most cases, vaccination is used as a preventive measure only in high-risk areas or for emergency immunization [2]. The primary reasons why many countries do not implement vaccination are summarized in Table 6 (data adapted from [72]).

The cost of vaccination is another primary concern for farmers. Manual vaccination of individual birds is time-consuming and labor-intensive, especially on large-scale farms, where multiple doses may be required [2]. In many cases, the cost of vaccination is covered by federal, state, or provincial governments; however, in some countries, the commercial sector bears the financial responsibility for both vaccines and administration costs. In the European Union, emergency vaccination costs are partially subsidized, with 100% of the vaccine and 50% of the vaccination costs covered by the EU [72].

#### 7.9.3. Additional Methods of Prevention

In addition to vaccines, a holistic approach to prevention and eradication is required. Vigilance and early reporting of HPAI outbreaks are critical in minimizing the impact and spread of the disease. However, one study found that the average outbreak management duration was 12.3 days from virus introduction to depopulation, primarily due to delays in HPAI reporting by farmers [118].

A higher level of farmer education, experience and training and larger farms with lower stocking densities correlate with increased biosecurity measures and decreased risk of AI outbreaks [92]. Several farm activities pose significant biosecurity risks and increase the likelihood of outbreaks, including bird transfers between farms, farm visitors, inadequate waste management, fomites, and other animal species on the farm [119].

Enhanced biosecurity measures include the following:Maintaining a closed flock or implementing quarantine for new flock members;Reducing stocking density;Preventing the presence of other animals on the farm;Implementing proper waste disposal practices;Limiting farm visitors and ensuring they do not introduce potential hazards into poultry houses, such as phones, jewelry, or paper handkerchiefs;Providing farmers and staff with accessible education and training on AI prevention.

Emphasizing the importance of early reporting to improve outbreak management is crucial.

## 8. Future Directions

The global spread of HPAI, particularly the H5N1 subtype, poses significant challenges to both animal and public health. Recent outbreaks have led to substantial economic losses in the poultry industry and have raised concerns about potential zoonotic transmission to humans. Vaccination could be a potential tool in controlling HPAI. However, evolving viral dynamics necessitate continuous evaluation, enhancement and potential implementation and expansion of vaccination strategies. The long-term PE of vaccines remains an under-researched area. While the short lifespan of most poultry may minimize the need for long-term immunity, this area still requires further investigation.

The ongoing H5N1 outbreaks in dairy cattle may necessitate the development of a bovine H5N1 vaccine if the stamping out protocol is not implementable in this species. In the long run, the introduction and routine use of poultry-specific H5N1 vaccines—particularly in developing countries—may become essential to mitigate severe losses in the poultry industry and support food security and sustainable agriculture.

While the reviewed studies provide valuable insights into vaccine efficacy under controlled laboratory conditions, validating these findings in real-world field environments is critical. Field studies are indispensable for understanding how biosecurity levels, farm management practices, stocking densities, and environmental conditions influence vaccine performance. Moreover, interactions between vaccinated and unvaccinated populations, viral transmission dynamics, and co-infections must be assessed to determine the practicality and scalability of vaccination protocols. Therefore, we recommend integrating field trials into the standard evaluation of vaccine efficacy to ensure their reliability and applicability under diverse farming conditions.

The following policy measures and funding priorities are proposed in subsections to enhance the effectiveness of AI control programs.

### 8.1. Establishing National and Regional Vaccination Protocols

Governments should develop and implement standardized vaccination guidelines practically tailored to the dominant viral strains in their respective regions. These protocols should provide clear recommendations regarding vaccine selection, administration schedules, and species-specific considerations to enhance vaccine efficacy and facilitate large-scale implementation.

The losses suffered by the farmers due to preventive measures (culling, trade restrictions, etc.) are higher, and the demand for vaccination is stronger even in countries practicing the “stamping out” policy. It is recommended that vaccination programs are drawn up at the national level by each country in a certain region. Still, it should be noted that infectious, contagious animal diseases do not respect national borders, and international trade (transport of live animals and animal products) further complicates the issue. Therefore, the animal health representatives of the countries in the region should coordinate and adopt an appropriate strategy for each member.

The appropriate vaccination strategy should be developed, taking into account the local epidemiological conditions and risk assessment. Vaccination is strongly recommended in areas frequently visited by wild birds, where large numbers of poultry holdings are present, and where large numbers of free-range large-scale poultry production is operating.

Vaccinated poultry species should be selected according to their economic value and growing/breeding period. The higher the economic value of a flock (poultry species) on a given poultry holding, the more vaccination is recommended. Furthermore, the longer the animals are kept, the stronger the recommendation is to introduce vaccination practices into the production system of the flock.

Animal welfare and economic considerations should be considered when carrying out vaccination. It is preferable to use hatchery vaccination, even in ovo, rather than manual vaccination on the farm. Species sensitivity is not negligible. The most sensitive poultry species—such as turkeys—should be favored over less sensitive species such as geese.

Research on vaccine effectiveness suggests that vector vaccines may be preferable to inactivated or other types of vaccines.

It is important to note that vaccination strategies and protocols need to be planned and implemented, taking into account the current international recommendations—the WOAH recommendation highlighted.

### 8.2. Investing in Field Trials

Investment in large-scale field trials is essential to validate vaccine efficacy under real-world conditions and inform policy decisions. These trials should reflect regional farming practices and account for diverse poultry species and breeds to ensure that vaccination strategies are adaptable to various production systems.

### 8.3. Development of Universal Avian Influenza Vaccines

Conventional vaccines are typically designed to target specific viral strains, which may limit their effectiveness against emerging variants. To address this challenge, research increasingly focuses on developing universal AI vaccines that elicit broad immune responses across multiple influenza subtypes. This approach aims to provide comprehensive protection while reducing the frequency of vaccine reformulation. Recent advancements in vaccine platforms, such as mRNA technology, offer promising avenues for the rapid development and deployment of universal vaccines, which could significantly enhance global preparedness for AI outbreaks [120].

### 8.4. Implementation of DIVA Strategies

A significant challenge in vaccination programs is distinguishing between infected and vaccinated animals. The development and implementation of DIVA-compatible vaccines, alongside corresponding diagnostic tests, are critical for enabling effective monitoring of viral circulation in vaccinated populations. These strategies are essential for maintaining surveillance efficacy and ensuring vaccination does not mask the presence of the virus and active infection [121]. DIVA vaccines could be particularly useful in surveillance zones and repopulated flocks, as the risk of emerging and re-emerging HPAI remains high. If the differentiation between infected and vaccinated birds is ensured, these vaccines could be crucial in disease control. Furthermore, the development of marker vaccines should go hand in hand with the advancement of discriminative ELISA tests, enabling reliable serological differentiation. Future research should prioritize the development of practical methods for implementing DIVA strategies in the field and the creation of marker vaccines to enhance their effectiveness. Moreover, promoting the adoption of DIVA-compatible vaccines can be facilitated through public-private partnerships and incentives for vaccine manufacturers. Supporting the widespread deployment of these vaccines would strengthen disease monitoring systems while preserving the integrity of international trade [122,123,124]. Emphasizing the importance of DIVA protocols and their practical application in real-world settings will be crucial to developing more effective and sustainable disease management strategies.

### 8.5. Integration of Vaccination with Comprehensive Control Measures

Vaccination should be incorporated into a complex control strategy that includes strict biosecurity measures, active surveillance, and the culling of infected flocks. Combining these approaches enhances the overall effectiveness of HPAI control programs [75]. For instance, the European Food Safety Authority emphasizes that vaccination should complement rather than replace other preventive and control measures [121].

### 8.6. Addressing Antigenic Drift and Shift

The high mutation rate of AI viruses can lead to antigenic changes, potentially reducing vaccine efficacy. Regular updates to vaccine strains, informed by ongoing surveillance data, are essential to maintaining vaccine effectiveness against circulating virus variants. This adaptive approach ensures that vaccination programs remain effective in an evolving viral landscape [125].

### 8.7. Enhancing Global Surveillance and Data Sharing

Effective vaccination strategies depend on robust global surveillance systems that monitor HPAI outbreaks and track viral evolution. Timely data sharing among countries and international organizations facilitates rapid responses to emerging threats and informs vaccine strain selection. Given the transboundary nature of AI, collaborative efforts are essential for disease control [126,127]. Strengthening surveillance systems requires increased funding to expand monitoring programs, detect emerging strains, and evaluate vaccine coverage and effectiveness. Additionally, real-time data sharing between countries is crucial for the rapid adaptation of vaccination strategies [126,128].

### 8.8. Enhancing Farmer Education and Subsidies

Farmers should receive targeted education programs to enhance their understanding of vaccination benefits, administration techniques, and biosecurity measures. Furthermore, financial subsidies or incentives should be introduced to alleviate the economic burden of vaccination, particularly for small-scale and resource-limited farmers [129].

Biosecurity is the basis of all disease prevention protocols [130], so it is essential that appropriate on-farm hygiene and preventive measures are implemented on poultry farms to reduce the risk of infection and the spread of infectious diseases. Hygiene and biosecurity are important not only for animal husbandry but also for the public health importance of AI.

### 8.9. Establishing Emergency Vaccine Banks

Developing and maintaining strategic vaccine reserves are essential for rapid deployment during outbreaks. This approach is particularly critical in regions with high migratory bird activity or areas prone to frequent HPAI outbreaks, ensuring that vaccines are readily available for immediate response efforts.

### 8.10. Public Health Preparedness and Human Vaccination

Given the zoonotic potential of HPAI viruses, preparing for possible human infections is crucial. Developing and stockpiling human vaccines against HPAI strains, particularly those with pandemic potential, is a proactive measure. For example, the U.S. government has allocated funding to develop mRNA vaccines targeting H5N1, aiming to enhance pandemic preparedness [131,132].

Advancing vaccination strategies against HPAI requires a multifaceted approach that includes the development of universal vaccines to provide broader protection against diverse HPAI strains, the implementation of DIVA strategies to facilitate disease surveillance and trade, and the integration of vaccination with comprehensive control measures such as biosecurity and outbreak containment. Additionally, adapting vaccines to viral evolution is essential to maintain their efficacy against emerging variants. Strengthening global surveillance is also a key component in ensuring early detection and rapid response to outbreaks. Furthermore, preparing for potential human transmission remains critical to safeguarding public health.

Addressing these key areas can more effectively integrate vaccination strategies into comprehensive AI control programs, ensuring economic sustainability for the poultry industry while protecting public health.

## 9. Conclusions

In conclusion, protecting large-scale poultry farms from AI—particularly the HPAI strain H5N1 clade 2.3.4.4b—requires a multi-faceted approach. While poultry may benefit from multiple doses of inactivated virus vaccines, a single dose might provide sufficient PE against clade 2.3.4.4b strains. However, ducks and geese may require multiple doses, and the long-term PE of these vaccines remains unknown. This highlights the clear need for large-scale field trials to validate vaccine efficacy under real-world conditions and for further research into the duration of protection offered by different vaccines across various poultry species. Vaccination remains a critical tool, and promising advancements in novel-technology vaccines offer hope for future containment. Nevertheless, challenges persist, including antigenic drift, vaccine compatibility, and the need for booster doses. Biosecurity measures—such as reducing flock density, controlling farm access, and improving waste management—are equally vital.

The development and implementation of DIVA protocols should be prioritized to enhance disease surveillance and facilitate international trade. In surveillance zones and repopulated flocks, where the risk of HPAI emergence and re-emergence remains high, the use of DIVA vaccines could provide a valuable tool for disease control. However, their effectiveness relies on accurately distinguishing infected individuals from vaccinated ones. To address this challenge, the advancement of discriminatory ELISA must accompany the development of marker vaccine tests that enable precise serological differentiation. Future research should focus on optimizing practical methodologies for field application of DIVA strategies and refine marker vaccines to improve their efficacy. The integration of DIVA protocols into disease control programs will not only reinforce trade security but also ensure more effective and transparent disease monitoring on a global scale. Educating farmers on early detection and proper farm hygiene can significantly reduce the risk of outbreaks.

To prevent future epidemics, collaborative efforts between governments, veterinarians, and the poultry industry will be essential in developing cost-effective, practical solutions, such as emergency vaccination banks. Focusing on the advancement of DIVA-compatible strategies and tools will enhance existing approaches, paving the way for sustainable disease management in the poultry industry. Prioritizing long-term studies on vaccine efficacy and protection duration will strengthen current strategies and provide the scientific foundation for more sustainable disease control in the poultry sector.

## Figures and Tables

**Figure 1 vaccines-13-00453-f001:**
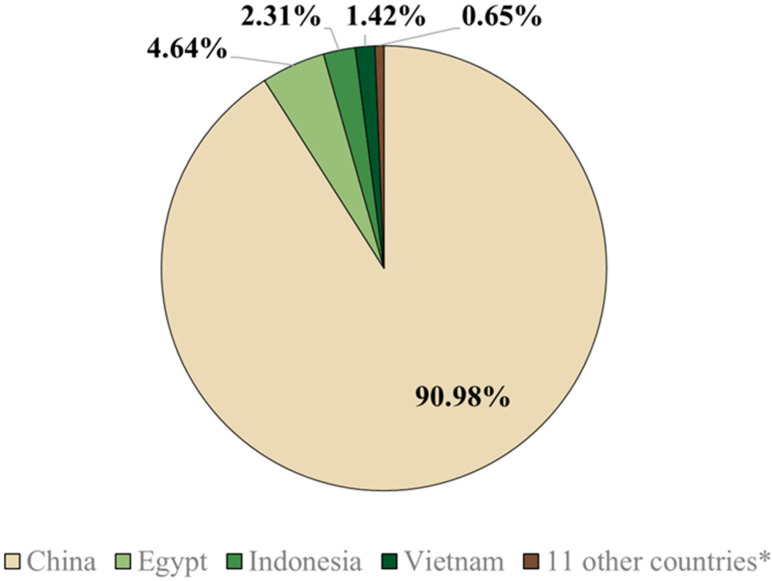
Percentages of the total vaccine doses administered against highly pathogenic avian influenza between 2002 and 2010, with a total of 115 billion doses. * 11 other countries represented include Côte d’Ivoire, France, Hong Kong, Israel, Kazakhstan, Mongolia, North Korea, Pakistan, Russia, Sudan, and the Netherlands [59].

**Figure 2 vaccines-13-00453-f002:**
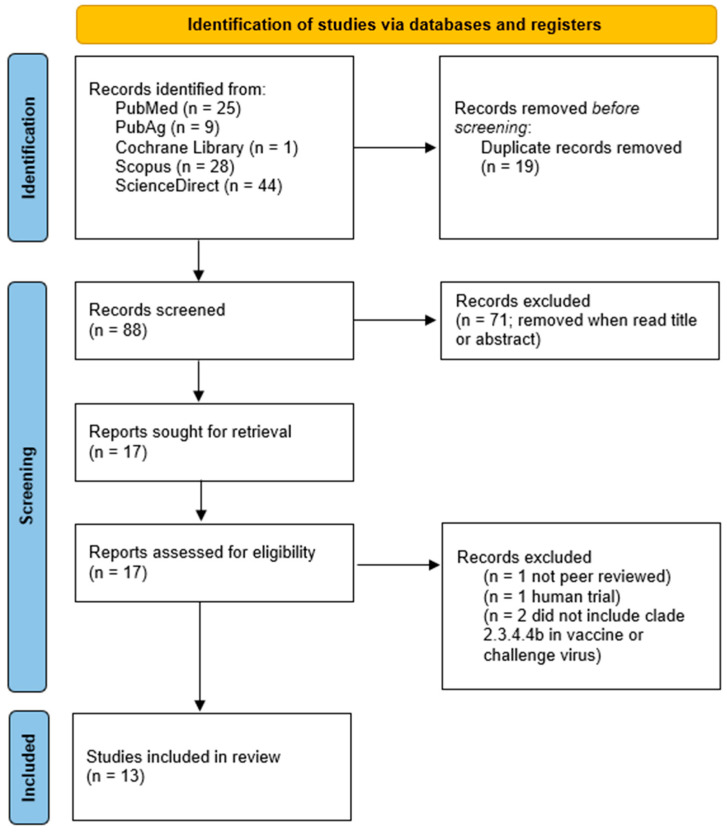
Results of the Boolean search “Clade 2.3.4.4b AND vaccine” in a Preferred Reporting Items for Systematic Reviews and Meta-Analyses (PRISMA) flowchart [97].

**Table 1 vaccines-13-00453-t001:** Clinical signs of avian influenza in poultry species, based on data from [9].

		LPAI	HPAI
Clinical signs	General	decreased feed and water consumption, lethargy, slight weight loss, diarrhea (occasionally)	found dead without prior clinical signs (in peracute disease), lethargy, recumbency, comatose state, decreased feed and water consumption
	Respiratory	sneezing, coughing, ocular and nasal discharge	less common than in LPAI
	Egg production	transient decrease in	cessation
	Neurological	-	varied symptoms (in acute to subacute cases)
	Mortality	low	high
Gross lesions	Respiratory	rhinitis, sinusitis, congested and edematous tracheal mucosa with occasional hemorrhages and luminal exudates. With secondary bacterial pathogens: fibrinopurulent bronchopneumonia, air sacculitis, and coelomitis	in peracute disease: no gross lesions, in acute disease: swelling (oedema) in the head, legs, and feet, subcutaneous hemorrhages, hemorrhages on serosal or mucosal surfaces, necrotic foci within multiple visceral organs, atrophic primary lymphoid organs, enlarged spleen with pale necrotic foci
	Egg production	ovaries regress, mature ova rupture, egg yolk peritonitis, swollen oviduct with luminal exudates, misshapen and thin-shelled eggs, eggs lack pigment	
	Urinary	swollen kidneys, visceral urate deposition (in hens)	
	Digestive	mild enteritis, pale, mottled pancreas with random hemorrhages (mainly in turkeys, rare)	
Histological lesions	Respiratory	heterophilic-to-lymphocytic rhinitis, sinusitis, tracheitis and bronchitis, interstitial pneumonia	more common than gross lesions, necrosis and/or inflammation with hemorrhages in multiple organs (especially within the skin, brain, heart, pancreas, lungs, adrenal glands, and lymphoid organs) due to viraemia and vascular damage
	Urinary	tubule necrosis, interstitial nephritis	
	Lymphoid	depletion	

HPAI—highly pathogenic avian influenza, LPAI—low pathogenic avian influenza.

## Data Availability

The data presented in this study are available within the article.

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
