# Peer review of "Avian Influenza Clade 2.3.4.4b: Global Impact and Summary Analysis of Vaccine Trials"

_vaccines, 2025, doi:10.3390/vaccines13050453_

Round 1

Reviewer 1 Report

Comments and Suggestions for Authors

This review highlights the importance of developing and evaluating vaccines against HPAI A(H5N1) clade 2.3.4.4b, a virus with significant implications for both animal and human health. By focusing on verifiable and reproducible studies, the review aims to provide reliable insights into vaccine efficacy, ultimately contributing to global efforts to control and prevent avian influenza outbreaks.The manuscript is well written and illustrated and may be accepted as it is.

Reviewer 2 Report

Comments and Suggestions for Authors

This detailed review summarizes advances and limitations for the use of H5N1 avian influenza vaccines in poultry, with focus on the dominant clade 2.3.4.4b. Various vaccine platforms are reviewed, both licensed and experimental. The authors conducted a thorough analysis of vaccine efficacy in experiments and on poultry farms, and pointed out factors that may influence vaccine efficacy in general, as well as farmers' willingness to vaccinate against HPAI viruses. The information presented here is important and will be of interest for wide audience. I have mainly comments on the layout of the text and tables.

Specific comments:

1.       Information on poultry vaccination against AI is outdated. Figure 1 shows data ending in 2010, which is almost 15 years ago. Given that the 2.3.4.4 HPAI became dominant only in 2014, this data should look different now.

2.       Tables 1, 2, 5 are difficult to read. Please add more dividing lines or organize the inserts so that they do not merge with each other. There is no need to divide into two major columns licensed and unlicensed vaccines in Table 2, and then add rows such as “where licensed”, as this is automatically not applicable for the unlicensed vaccines. Please reorganize this table for better reading.

3.       The footnotes for the tables should be provided below the table, not within their titles.

4.       Is subsection 3.1 really necessary? There is no other subsection in section 3.

5.       The abbreviation HPAI is only deciphered on line 74 and is used for the first time on line 69. Same with AB: first mentioned on line 358, but deciphered on line 369.

6.       Line 107: antigenic drift occurs via point mutations, not shift.

7.       Lines 243-247: Please list specific vaccines that are licensed by USDA first, and then provide information on the producers.

8.       Line 298: “All vaccine trials … performed their studies” – please revise the sentence as trials cannot perform studies.

9.       Line 302: “another a hybrid” – please correct

10.   The sentences In lines 296-297 and 306-307 are exactly the same, please revise.

11.   Table 5: “Trivalent activated 2.2.1.1 and 2.2.1.2.” – really activated? Or inactivated?

12.   Table 5: “decreased only 1/15 birds dpc” – what does “dpc” stand for? How many days post challenge?

13.   Line 365: “showed titers above HI titers 5 log2” please revise.

14.   Line 357: “A higher HI titer correlates with lower mortality and reduced shedding”, line 373: “Higher HI titer correlates with reduced viral shedding in most studies as in Table 5” – please avoid such repetitions.

15.   Lines 374-376: “viral shedding via oral or fecal route was measured by samples from tracheal and/or cloacal swabs after challenge. Viral shedding via fecal or oral route was measured by samples with tracheal and/or cloacal swabs after being challenged with the vaccine.” – seems like saying the same thing twice.

Comments on the Quality of English Language

there are frequent repetitions and incoherent sentences in the text

Reviewer 3 Report

Comments and Suggestions for Authors

L62-63: While you have described some of the mammals, I would also recommend including others such as foxes, bears, seals, dogs, goats, zoo animals (tigers and leopards) etc. to highlight the widespread detection of the virus.

L112-117: The commonly accepted idea of "pig as a mixing vessel" for AIV has recently been challenged. I would highly recommend that the authors explore this concept and may be argue for or against this fundamental idea. Some of the publications include the following: 

- Hennig, C., Graaf, A., Petric, P. P., Graf, L., Schwemmle, M., Beer, M., & Harder, T. (2022). Are pigs overestimated as a source of zoonotic influenza viruses?. Porcine Health Management, 8(1), 30.

- https://wwwnc.cdc.gov/eid/article/30/4/23-1141_article?utm_source=chatgpt.com

L124: "is assumed"

L119-148: This section looks more like a report. How is this related to vaccine efficacy? I would suggest to the authors that they replace this with another section that provides some background information on vaccines. 

L177-178: This may not be true for all AI viruses. For instance, data from the 1918 pandemic shows that younger adults were more affected compared to children. This has be speculated to be due to cytokine storm. Please consider all aspects and contexts to what you put as a factual statement. 

L191-195: Please be specific with numbers. This section does not provide sufficient information. For instance, you can mention the total number of birds that have died (including from culling) as of, say January 2025, instead of mentioning "per farm loss", which is not useful. 

L197: Specify which part of the world farmers can keep their chickens, cull their chickens and sell their chickens.

 L197: Avoid using terms such as "many countries". As a scientific review paper, it needs to be specific. 

L215-219: The statement makes sense and highlights a valid perspective but oversimplifies the complex web of factors influencing vaccination reluctance. The authors are missing some key factors/reasons for not vaccinating poultry, such as;

  • Economic and Trade Considerations:

    • The reluctance to vaccinate is often driven by trade restrictions. Countries importing poultry products may prohibit imports from regions that use HPAI vaccines because it complicates distinguishing between infected and vaccinated birds.
  • Vaccine Efficacy and Logistics:

    • Vaccines may not fully protect against all emerging HPAI variants, especially if the virus mutates rapidly. Logistical challenges in vaccinating large poultry populations also play a role.
  • Ethical and Practical Considerations:

    • Relying on non-vaccinated poultry as sentinels could be seen as unethical, especially if effective vaccines are available to prevent suffering and mass culling.
  • Emerging Trends:

    • Advances in vaccines (e.g., DIVA vaccines—Differentiating Infected from Vaccinated Animals) are addressing some concerns, potentially reducing the reliance on non-vaccinated poultry as sentinels.

L223: "115 billion vaccines" needs a unit

L254-256: Add a total volume of vaccine to Figure 1. % means nothing without having a context of total volume. 

L263-265: This needs to be updated with a recent data. What you've provided is from 2022, and now we have approached close to 150 million birds. 

Before moving to next sections, I highly suggest that the introduction section (above this point) focus on vaccines, instead of some general topics about transmission. At least more focus should be on background information on vaccines. 

L582: "On the long run introduction" should be changed to "In the long run introduction".

More detailed comments: 

    • The need for large-scale field trials to validate vaccine efficacy should be a prominent point in the conclusion.
    • The conclusion should clearly call for further research into the long-term protection offered by different vaccines, and for studies that monitor the duration of protection in different poultry species. 
    • While the review calls for the use of DIVA protocols, it could stress the importance of developing and implementing these strategies to allow for more effective monitoring and ensure the maintenance of international trade. It should be suggested that future research should focus on developing practical methods for implementing DIVA in the field, as well as the creation of marker vaccines.
    • Since co-circulation of different strains is a concern, the review could benefit from a more detailed discussion of the advantages of multi-valent vaccines. The review could further investigate the potential for antigen interference when using multivalent vaccines.
    • In the abstract, the review states that AI can cause mortality rates approaching 100%. However, later in the text, it is mentioned that LPAI typically has a mortality rate of <5%. It should be clarified that the high mortality rate is specific to HPAI, while LPAI has a lower mortality rate.
    • The review states that individual HI titers greater than 3 log2 were considered positive, but also notes that the WOAH stated that 4 log2 is a positive result. This discrepancy could be confusing for readers. It should be clarified what the standard for a positive HI titer is according to the WOAH, and that a titre of 3 log2 was used in the research reviewed in this paper.
    • The review makes the claim that "most of the vaccines provided nearly 100% protection against clinical signs in vaccinated groups". However, this is an overstatement. For example, Azab et al. found that 20% of birds showed clinical signs when challenged with H5N8 after being vaccinated with one vaccine, and 80% showed clinical signs when vaccinated with another. It should be noted that some vaccines do not offer 100% protection.

Reviewer 4 Report

Comments and Suggestions for Authors

The current H5N1 (2.3.4.4b) avian influenza panzootic poses significant threats to poultry production, wildlife, and human health. This manuscript describes epidemiological aspects, pathogenicity, clinical symptoms and lesions, public health relevance and economic significance of HPAI caused by influenza A  (H5N1) clade 2.3.4.4b virus with a focus on the efficacy of vaccines against HPAI. The information sounds useful for relevant researchers and policy-makers. Below are my comments.

1.  Lack of relevant global epidemiological data about influenza A  (H5N1) clade 2.3.4.4b virus, potentially leading to shadow this review’s significance.   

2. Scientific inaccuracy in some parts of the manuscript

·         Lines 298-299: are there any publications regarding trials in commercial chickens?

Line 381: 4.6. Maternally derived Abs. I feel a bit confused about maternal antibodies from SPF chickens. SPF chickens should be free of AI. Did it refer to commercial chickens, which was not in line with the description in Lines 298-299 ? 

·         Line 338: Mortality: this is an indicator of experimental vaccines’ effectiveness. Despite the summary in Tables 4 and 5. It is recommended that provision of detailed information about each vaccine in this section.

Likewise, Line 372: 4.5. (Virus) Shedding is an important indicator for the assessment of the vaccine efficacy. This section would require a bit detailed information about virus shedding  for each vaccine. Table 5: Did (Virus) Shedding quantify or provide a bit detail, if applicable?

·         Table 4: As the dose of challenge viruses is critical to assess the efficacy of experimental vaccines, addition of doses could be useful. Replacing “Author, Year” with “ref ” is recommended. Addition of vaccine strain clades if applicable would be better.

·         Line 356: As antigens used impact HI titres of specimens, specific antigens should be mentioned.

·         Lines 490-493: Detailed information is recommended.

·         Lines 330-333: provision of a bit background information about the vaccines may be better for readers to understand, although it was summarised in Tables 4 and 5.  

·         Line 410: which clade did A H5 vaccine belong to?

3. Further revision of this manuscript writing is required towards publication as some expressions are inaccurate, inappropriate or not logically developed. For example,  

·         influenza in Line 52 (should be “avian influenza”),

·         Lines 114-117 (. Although…) ,

·         Line 120 (Another possible means of.. ) ,

·         Table 1: to be modified for the readers to easily followed.  

·         Line 261: Is the subtitle necessary?

·         Lines 293-295: it could be better to make it simpler.

·         Lines 449-455: Limitation of using inactivated vaccines needs to be added.   

·         Line 414: The sentence “Usually, a single dose of AI vaccine is given.” requires revision.  

·         Lines 419-422 : This did not make sense.

·         Lines 431: grammar error

·         Line 510: not properly expressed.

·         Line 573: Abbreviation HPAI should be mentioned earlier in the manuscript.

Comments on the Quality of English Language

Must be improved towards publication 

Round 2

Reviewer 2 Report

Comments and Suggestions for Authors

The authors have significantly revised the article according to the reviewers' comments. This now is suitable for publication in Vaccines

Author Response

Comments 1: The authors have significantly revised the article according to the reviewers' comments. This now is suitable for publication in Vaccines.

Response 1: Thank you for your thoughtful and encouraging feedback on our manuscript. We are especially pleased that you find the manuscript suitable for acceptance in its current form. All the authors would like to thank you once again for your time and expertise in reviewing our work.

Reviewer 3 Report

Comments and Suggestions for Authors

No comment

Author Response

Comments 1: No comment.

Response 1: Thank you very much for your valuable comments and for taking the time to review our manuscript.

Reviewer 4 Report

Comments and Suggestions for Authors

The authors have made great efforts to revise the manuscript. Tables 4 and 5 seemed improved.  However, the review paper is not simply the accumulation of scientific data. It needs structuring your points in a clear and concise manner. Some sections may not be necessary or cumbersome. To this end, I would like to see a concise and clear version by re-organising the paragraphs or contents in a logical order. Moreover, language barrier appeared another issue for improvement.

For example, sections 2.4 and 2.5 could be moved to an earlier section of this manuscript.  Section 5.6.3 could be combined with Table 3. Lines 439-445: you are talking about the protection of vaccine using HI tire cut-off rather than using positive cut-off of HI titre. Using 2 or 3 sentences is sufficient to present your point.

Comments on the Quality of English Language

The paper needs to be written in a clear and concise manner. 

Author Response

Response to Reviewer 4 Comments

Comments 1: The authors have made great efforts to revise the manuscript. Tables 4 and 5 seemed improved.  However, the review paper is not simply the accumulation of scientific data. It needs structuring your points in a clear and concise manner. Some sections may not be necessary or cumbersome. To this end, I would like to see a concise and clear version by re-organising the paragraphs or contents in a logical order.

Response 1: Thank you for pointing out this issue. We agree with this comment. Therefore, we have revised and reorganized the text.

In line 45-233.:

“1. Introduction

The term influenza originally referred to epidemics of acute, rapidly spreading catarrhal fevers in humans caused by viruses belonging to the Influenzavirus A genus of the Orthomyxoviridae family [1]. As an enveloped virus with a segmented RNA genome, it can readily alter its antigenic properties through mutation and genetic reassortment. The evolving nature of this virus necessitates the continuous development of new and effective vaccines to combat emerging strains. Additionally, its potential adaptation to new hosts heightens concerns about the risk of a zoonotic pandemic [2].

Orthomyxoviruses are recognized as the cause of significant numbers of natural infections and diseases, usually of the upper respiratory tract. Cases are commonly observed in humans, horses, dogs, domestic pigs, and various bird species. Sporadic infections have also been reported in mink and several marine mammals [3–5]. Recently, a fast-spreading epidemic caused by the highly pathogenic avian influenza (HPAI) A(H5N1) clade 2.3.4.4b virus was reported among dairy cattle in the United States [6].

This review aimed to identify all relevant publications on HPAI A(H5N1) clade 2.3.4.4b vaccines, especially the testing of their efficacy. For this analysis, the Boolean search term "Clade 2.3.4.4b AND vaccine" was used in the following databases: PubMed, PubAg, Scopus, Cochrane Library, and ScienceDirect. Only experiments conducted in an appropriate, verifiable, and reproducible manner were included, thereby increasing the reliability of the conclusions drawn. Inclusion criteria were clinical trials or randomized controlled trials evaluating vaccines against HPAI clade 2.3.4.4b in chickens, ducks, or geese. Papers were included if they measured viral shedding, hemagglutination inhibition (HI) titers, mortality, and morbidity. Additionally, studies had to include clade 2.3.4.4b as either the vaccine strain or the challenge virus. Multiple studies examined more than one vaccine; in such cases, only the results for clade 2.3.4.4b vaccines or vac-cine groups tested against clade 2.3.4.4b were considered.

2. General characteristics of avian influenza

Avian influenza (AI) is one of the most devastating poultry diseases, first identified in the 19th century. It is a viral disease caused by influenza type A viruses, affecting the respiratory, digestive, and nervous systems of multiple bird species, including domestic poultry and wild aquatic birds [7]. Infection in domestic poultry can result in various clinical outcomes (Table 1). The low pathogenic avian influenza (LPAI) biovariants generally cause asymptomatic infections or mild symptoms, such as respiratory disease and reduced egg production. At the same time, HPAI often leads to severe systemic disease with mortality rates reaching 100% [8]. It is commonly observed in gallinaceous birds [9] but has also been reported in domestic aquatic birds [10]. However, wild aquatic birds, such as ducks and geese, are more likely to serve as a reservoir for LPAI strains [9]. Wild waterfowl can shed the virus asymptomatically [5], challenging surveillance and eradication efforts. Since HPAI strains originate from LPAI strains [11,12], the spread of HPAI from LPAI-infected sources remains a significant concern. Moreover, influenza A viruses can undergo genetic changes through antigenic drift and anti-genic shift. Antigenic drift occurs through point mutations in the viral genome, leading to minor changes in the virus. On the other hand, the antigenic shift involves genetic reassortment, resulting in significant viral changes that enable the virus to cross species barriers and circulate in new populations [13].

2.1. AI in mammalian species

Although AI was historically considered an avian disease and hosts of HPAI include a wide variety of birds [14], infections have been documented in various mammalian species, including cats, pigs, ferrets, mice, macaques, cattle, foxes, beavers, bears, seals, dogs, goats, tigers, and leopards, as well as humans [15–20]. The USDA has begun assessing the potential to develop an effective vaccine against the H5N1 bird flu virus in cattle. However, it has cautioned that it is too early to determine how long the process will take. The virus has infected multiple dairy herds across several U.S. states [21].

As a result of antigenic shift, avian-to-mammalian transmission can occur, with pigs serving as an intermediate host, as they possess both α2,6 and α2,3 receptors for influenza viruses, similar to birds [22]. Suppose a pig is simultaneously infected with a mammalian and an AI virus. In that case, gene reassortment may occur, potentially leading to the emergence of a novel virus capable of human transmission [23]. However, it is essential to note that pigs have demonstrated low susceptibility to H5N1 clade 2.3.4.4b [24]. While this mechanism suggests a potential role for pigs as mixing vessels in AI epidemiology, the question remains unsettled. Conflicting scientific opinions exist regarding the significance of pigs in this process, and further studies are needed to clarify their exact role [25].

Given that HPAI A(H5N1) has recently infected more mammalian species than ever before [19] and HPAI A(H5N1) clade 2.3.4.4b viruses have acquired a significant number of mammalian adaptation mutations in recent years [6,26,27] the risk of its turning into a human pathogen is growing. Reports of AI infections in humans through direct contact with infected animals have increased over the past decade [28], with some cases proving fatal [18]. One source of exposure is wet markets or live poultry markets, and the closure of these places has been associated with a greater prevalence of AI in humans [29,30]. Farmers and poultry workers are advised to take stringent measures to prevent infection [15,31–33]; however, in a minority of cases, humans have contracted HPAI without any known exposure to poultry [29].

2.2. Public health relevance

Certain HPAI strains are zoonotic and capable of causing infections in humans. It has been acknowledged that AI was likely the source of human influenza pandemics [34] in the past and has the potential to trigger future ones [35]. This is particularly true given that HPAI A(H5N1) has recently infected more mammalian species than ever [19]. Furthermore, HPAI A(H5N1) clade 2.3.4.4b viruses have acquired a significant number of mammalian adaptation mutations in recent years [6,26,27]. A potential route of AI transmission involves direct avian-to-human transmission, followed by subsequent human-to-human spread [23]. Most reports of cases of AI in humans have claimed the infected person was in contact with birds either in wet bird markets [36,37] or farms [15,38]. However, human-to-human transmission has also been reported [39] as limited and generally non-sustained. It has been observed in small clusters or groups [40] and is assumed to pose a low risk presently [31,41].

Clinical signs in humans include fever, respiratory symptoms, vomiting, diarrhea, and, in severe cases, death [32,33]. Children, in particular, are at a high risk of death from H5N1 [13]. Recommended protective measures include wearing appropriate protective clothing when handling poultry, implementing health monitoring systems for human populations, and vaccination. However, the current vaccine supply is insufficient to meet the probable demand during a pandemic. Prophylactic and therapeutic treatments, such as adamantanes and the newer class of neuraminidase (NA) inhibitors, are also available [13].

2.3. Transmission

In humans, influenza viruses can be transmitted through the air and direct contact with an infected host. Airborne transmission is facilitated by temperate weather with low humidity, which aligns with the general perception of “flu season” [42]. Although AI is an influenza virus, human-to-human airborne transmission has not been observed thus far. However, airborne transmission without an intermediate host has occurred in mammals such as ferrets [43] and cattle [6]. In poultry, a study by Zhao et al. [44] suggested that airborne transmission could seriously impact farm-to-farm transmission. However, the airborne transmission of HPAI clade 2.3.4.4b between farms was shown to be limited in experimental settings [45] and, more recently, in the field, where James et al. [46] reported that the virus travelled only short distances (10-80 m). The probability of airborne infection is thought to be influenced by the farm type, housing type, flock size, flock density, and the number of infected animals [47]. A decreased flock density and improved husbandry, biosecurity, and ventilation measures could help limit airborne transmission [44].

Far more effective than airborne transmission, contact transmission is thought to be responsible for most cases of AI [47]. Non-airborne transmission sources and fomites, such as contaminated feathers, spread to 80m [46]. Therefore, wild birds and mammals pose a risk to farm birds as they can enter farms and spread the virus via direct contact with farm animals or contaminate food sources. Surveillance and prevention of wild animal exposure may be helpful to reduce this risk [48]. In addition, indoor farming can help to reduce the risk of contamination via the droppings of migratory birds, which sometimes flock in great numbers.

Transmission of AI from wild to domestic birds occurs more frequently than the reverse, but dispersal among similar groups (e.g., wild-to-wild or domestic-to-domestic) is most efficient [49].

3. Pathogenicity, clinical symptoms and lesions in poultry

AI is commonly classified into LPAI and HPAI [50]. LPAI is typically associated with localized infections in the respiratory and alimentary tracts and generally has a low mortality rate (<5%) [9]. Its primary impact on the poultry industry relates to flock welfare, decreasing egg and meat production [9,51,52]. In contrast, HPAI had a severe effect on the poultry industry [53–55], causing systemic infections and high mortality rates. It can lead to respiratory disease, hemorrhages, necrosis, and inflammation of multiple organs [9]. Some possible clinical signs and lesions are listed in Table 1.

Gallinaceous species are highly susceptible to HPAI. Severe clinical disease and death develop rapidly after infection. Mortality can reach 100% in affected large-scale holdings within a few days. Domestic waterfowl often exhibit a more protracted course of disease. The clinical outcome might be mild or even pass unnoticed, depending on the circulating virus strain. However, in some cases, infection can be devastating. The disease severity and mortality among wild birds can vary significantly [2,50,51]. The out-come of the infection depends on multiple factors, including viral adaptation to the host, host immunogenicity, affected bird species, and the specific strain involved [9]. AI strains are classified based on their hemagglutinin (HA) and neuraminidase (NA) surface proteins [50]. Since influenza type A viruses continuously mutate through antigenic drift [13], HPAI can emerge from LPAI through intrinsic mutations in poultry hosts [56]. Currently, HPAI mutations have only been observed in LPAI strains of H5 and H7 HA subtypes, as all known HPAI viruses belong to either the H5 or H7 subtypes, whereas LPAI strains can belong to any of the 18 HA subtypes [50]. The exact factors driving this seemingly spontaneous shift remain unknown [11,29]. Both H5 and H7 strains have been detected in chickens[57] and in humans [35,57], making them strains of concern for both the poultry industry [51,58] and public health [35].

4. Economic significance of AI

AI is responsible for the deaths of millions of birds due to disease or culling [59], making it a significant economic burden for poultry farmers [53,54,60]. The virus is widespread globally, with outbreaks reported in Europe [61,62], Asia [63], North America [64,65], Africa [66,67], and Australia [68].

Over the past two decades, AI outbreaks have sharply increased, leading to devastating economic losses in the poultry industry. Between 2005 and 2020, outbreaks resulted in the deaths of 246 million birds worldwide [2,69]. Since January of 2022, more than 130 million birds have been affected in the United States alone [70], along with approximately 3,500 outbreaks reported in Europe between 2020 and 2024 [71]. These outbreaks have incurred billions of USD in costs due to the direct impacts of infection and culling, trade restrictions, and the adverse effects on public perception [7].

Once a flock is proven to be infected with HPAI and the birds are therefore culled, nearby farms have several options, depending on the country or region. These options include enhanced biosecurity and establishing quarantine, culling the entire flock with financial compensation, or vaccinating flocks in the area [72]. In Germany, Australia, the Netherlands, Belgium, the United States, China, Ethiopia, Thailand, Senegal, Mauritania, and other countries, farmers may be compensated for up to 50% of the value of a healthy bird [54,72–74] with funds provided by the government or through government-industry partnerships [2,74]. However, while compensation helps mitigate economic losses, it may not be sufficient to incentivize farmers to cull their flocks [72]. In some developing countries where farmers receive no compensation, selling infected birds to live markets may be seen as a way to reduce financial losses. However, this practice poses serious risks for epidemic prevention.

The economic impact of AI on poultry farming increases the incentive to implement biosecurity and other preventive measures, which can reduce the spread of the disease [54,73,75].

AI causes instability in the poultry sector. High mortality rates, reduced production, and declining consumer demand for poultry products result in significant income losses, forcing some farmers to abandon poultry farming in search of alternative sources of income [53]. Since rural livelihoods and food security depend on the eradication of HPAI, controlling the disease is of paramount importance.

Additional costly factors include expenses related to disease prevention [76,77] and the cost of medical treatment in cases of human infection [78].

All these factors led to the introduction of vaccination in developing countries, and even countries practicing the “stamping out” policy are sometimes under pressure from the poultry industry to work out vaccination strategies for their poultry stocks in case of emergencies [59].”

We also added structured and clear text to form a more comprehensive manuscript.

In line 754-778 “It is recommended that vaccination programs are drawn up at the national level by each country in a certain region. Still, it should be noted that infectious, contagious animal diseases do not respect national borders, and international trade (transport of live animals and animal products) further complicates the issue. Therefore, the animal health representatives of the countries in the region should coordinate and adopt an appropriate strategy for each member.

The appropriate vaccination strategy should be developed, taking into account the local epidemiological conditions and risk assessment. Vaccination is strongly recommended in areas frequently visited by wild birds, where large numbers of poultry holdings are present, and where large numbers of free-range large-scale poultry pro-duction is operating.

Vaccinated poultry species should be selected according to their economic value and growing/breeding period. The higher the economic value of a flock (poultry species) on a given poultry holding, the more vaccination is recommended. Furthermore, the longer the animals are kept, the stronger the recommendation is to introduce vaccination practices into the production system of the flock.

Animal welfare and economic considerations should be considered when carrying out vaccination. It is preferable to use hatchery vaccination, even in ovo, rather than manual vaccination on the farm. Species sensitivity is not negligible. The most sensitive poultry species—such as turkeys—should be favored over less sensitive species such as geese.

Research on vaccine effectiveness suggests that vector vaccines may be preferable to inactivated or other types of vaccines.

It is important to note that vaccination strategies and protocols need to be planned and implemented, taking into account the current international recommendations—the WOAH recommendation highlighted.”

And in line 746-778.

“8.1. Establishing national and regional vaccination protocols

Governments should develop and implement standardized vaccination guidelines practically tailored to the dominant viral strains in their respective regions. These protocols should provide clear recommendations regarding vaccine selection, administration schedules, and species-specific considerations to enhance vaccine efficacy and facilitate large-scale implementation.

The losses suffered by the farmers due to preventive measures (culling, trade restrictions, etc.) are higher, and the demand for vaccination is stronger even in countries practicing the “stamping out” policy. It is recommended that vaccination programs are drawn up at the national level by each country in a certain region. Still, it should be noted that infectious, contagious animal diseases do not respect national borders, and international trade (transport of live animals and animal products) further complicates the issue. Therefore, the animal health representatives of the countries in the region should coordinate and adopt an appropriate strategy for each member.

The appropriate vaccination strategy should be developed, taking into account the local epidemiological conditions and risk assessment. Vaccination is strongly recommended in areas frequently visited by wild birds, where large numbers of poultry holdings are present, and where large numbers of free-range large-scale poultry production is operating.

Vaccinated poultry species should be selected according to their economic value and growing/breeding period. The higher the economic value of a flock (poultry species) on a given poultry holding, the more vaccination is recommended. Furthermore, the longer the animals are kept, the stronger the recommendation is to introduce vaccination practices into the production system of the flock.

Animal welfare and economic considerations should be considered when carrying out vaccination. It is preferable to use hatchery vaccination, even in ovo, rather than manual vaccination on the farm. Species sensitivity is not negligible. The most sensitive poultry species—such as turkeys—should be favored over less sensitive species such as geese.

Research on vaccine effectiveness suggests that vector vaccines may be preferable to inactivated or other types of vaccines.

It is important to note that vaccination strategies and protocols need to be planned and implemented, taking into account the current international recommendations—the WOAH recommendation highlighted.”

Comments 2:  Moreover, language barrier appeared another issue for improvement.

Response 2: Thank you for pointing out this issue. We agree with this comment. Therefore, we revised and corrected the manuscript with the help of a native speaker proofreader.

Comments 3: For example, sections 2.4 and 2.5 could be moved to an earlier section of this manuscript.

Response 3: Thank you for pointing out this issue. We agree with this comment. Therefore, we corrected this two section and moved forward with the other modification of sections which can be found in Comment 1.

In line 120-139: “2.2. Public health relevance

Certain HPAI strains are zoonotic and capable of causing infections in humans. It has been acknowledged that AI was likely the source of human influenza pandemics [34] in the past and has the potential to trigger future ones [35]. This is particularly true given that HPAI A(H5N1) has recently infected more mammalian species than ever [19]. Furthermore, HPAI A(H5N1) clade 2.3.4.4b viruses have acquired a significant number of mammalian adaptation mutations in recent years [6,26,27]. A potential route of AI transmission involves direct avian-to-human transmission, followed by subsequent human-to-human spread [23]. Most reports of cases of AI in humans have claimed the infected person was in contact with birds either in wet bird markets [36,37] or farms [15,38]. However, human-to-human transmission has also been reported [39] as limited and generally non-sustained. It has been observed in small clusters or groups [40] and is assumed to pose a low risk presently [31,41].

Clinical signs in humans include fever, respiratory symptoms, vomiting, diarrhea, and, in severe cases, death [32,33]. Children, in particular, are at a high risk of death from H5N1 [13]. Recommended protective measures include wearing appropriate protective clothing when handling poultry, implementing health monitoring systems for human populations, and vaccination. However, the current vaccine supply is insufficient to meet the probable demand during a pandemic. Prophylactic and therapeutic treatments, such as adamantanes and the newer class of neuraminidase (NA) inhibitors, are also available [13].”

In line 197-233: 4. Economic significance of AI

AI is responsible for the deaths of millions of birds due to disease or culling [59], making it a significant economic burden for poultry farmers [53,54,60]. The virus is widespread globally, with outbreaks reported in Europe [61,62], Asia [63], North America [64,65], Africa [66,67], and Australia [68].

Over the past two decades, AI outbreaks have sharply increased, leading to devastating economic losses in the poultry industry. Between 2005 and 2020, outbreaks resulted in the deaths of 246 million birds worldwide [2,69]. Since January of 2022, more than 130 million birds have been affected in the United States alone [70], along with approximately 3,500 outbreaks reported in Europe between 2020 and 2024 [71]. These outbreaks have incurred billions of USD in costs due to the direct impacts of infection and culling, trade restrictions, and the adverse effects on public perception [7].

Once a flock is proven to be infected with HPAI and the birds are therefore culled, nearby farms have several options, depending on the country or region. These options include enhanced biosecurity and establishing quarantine, culling the entire flock with financial compensation, or vaccinating flocks in the area [72]. In Germany, Australia, the Netherlands, Belgium, the United States, China, Ethiopia, Thailand, Senegal, Mauritania, and other countries, farmers may be compensated for up to 50% of the value of a healthy bird [54,72–74] with funds provided by the government or through government-industry partnerships [2,74]. However, while compensation helps mitigate economic losses, it may not be sufficient to incentivize farmers to cull their flocks [72]. In some developing countries where farmers receive no compensation, selling infected birds to live markets may be seen as a way to reduce financial losses. However, this practice poses serious risks for epidemic prevention.

The economic impact of AI on poultry farming increases the incentive to implement biosecurity and other preventive measures, which can reduce the spread of the disease [54,73,75].

AI causes instability in the poultry sector. High mortality rates, reduced production, and declining consumer demand for poultry products result in significant income losses, forcing some farmers to abandon poultry farming in search of alternative sources of income [53]. Since rural livelihoods and food security depend on the eradication of HPAI, controlling the disease is of paramount importance.

Additional costly factors include expenses related to disease prevention [76,77] and the cost of medical treatment in cases of human infection [78].

All these factors led to the introduction of vaccination in developing countries, and even countries practicing the “stamping out” policy are sometimes under pressure from the poultry industry to work out vaccination strategies for their poultry stocks in case of emergencies [59].”

Comments 4: Section 5.6.3 could be combined with Table 3. 

Response 4: Thank you for pointing out this issue. We agree with this comment. Therefore, we change Table 3 and added some additional information in line 357-368.

Comments 5: Lines 439-445: you are talking about the protection of vaccine using HI tire cut-off rather than using positive cut-off of HI titre. Using 2 or 3 sentences is sufficient to present your point.

Response 5: Thank you for pointing out this issue. We agree with this comment. Therefore, we change this section in line 439-444. “HI, titer and virus shedding results can be seen in Table 5. The cut off values of HI titers differ depending on the amount of virus applied in the tests. HI, titers greater than 3 log₂ were considered positive in certain studies [101,102] at a dilution of ≥4 log2 against 4 hemagglutinating units (HAU) of antigen, according to WOAH standards. However, some laboratories use 8 HAU in HI assays, which is an acceptable modification, but this affects the cut-off level, and a positive titer is defined as ≥3 log2 [56].”

4. Response to Comments on the Quality of English Language

Point 1: The paper needs to be written in a clear and concise manner.

Response 1: Thank you for your comment on our manuscript. We have reread the text multiple times and revised the entire manuscript to make it clearer and more concise for the readers.

5. Additional clarifications

-